# On the Calibration of Multiclass Classification with Rejection

**Chenri Ni[1]**   **Nontawat Charoenphakdee[1,2]**   **Junya Honda[1,2]**   **Masashi Sugiyama[2,1]**

[1] The University of Tokyo, Japan    [2] RIKEN Center for Advanced Intelligence Project, Japan

{nichenri, nontawat}@ms.k.u-tokyo.ac.jp
{jhonda, sugi}@k.u-tokyo.ac.jp

## Abstract

We investigate the problem of multiclass classification with rejection, where a classifier can choose not to make a prediction to avoid critical misclassification. First, we consider an approach based on simultaneous training of a classifier and a rejector, which achieves the state-of-the-art performance in the binary case. We analyze this approach for the multiclass case and derive a general condition for calibration to the Bayes-optimal solution, which suggests that calibration is hard to achieve by general loss functions unlike the binary case. Next, we consider another traditional approach based on confidence scores, in which the existing work focuses on a specific class of losses. We propose rejection criteria for more general losses for this approach and guarantee calibration to the Bayes-optimal solution. Finally, we conduct experiments to validate the relevance of our theoretical findings.

## 1   Introduction

In real-world classification tasks, e.g., medical diagnosis, autonomous driving, and product inspection, misclassification can be costly and even life-threatening. Classification with rejection is a framework aiming to prevent critical misclassification by providing an option not to make a prediction at the expense of the pre-defined rejection cost [6, 7]. If the rejection cost is less than the misclassification cost, there is an incentive to reject an instance. In practice, once the reject option is selected, one may gather more information about the instance or ask experts to give the correct label.

Much research on the theoretical perspective of classification with rejection has been devoted to the binary classification scenario [16, 1, 14, 29, 8, 9]. However, rather less attention has been paid to the multiclass scenario, which is undoubtedly important for real-world applications and is a more general framework. To the best of our knowledge, although there exist many methods that rely on heuristics [11, 28, 23], only the work by Ramaswamy et al. [20] provides the theoretical guarantee for their method. Nevertheless, the work by Ramaswamy et al. [20] only focuses on specific types of non-differentiable losses and their method requires re-training of the classifier when the rejection cost changes.

The key concept to validate the soundness of the method for classification with rejection lies in the notion of *calibration*, i.e., *infinite-sample consistency* [29, 8]. Calibration suggests that the minimizer of a surrogate risk behaves identically to the Bayes-optimal solution almost surely. The existing methods with calibration guarantees can be divided into two categories, which we detail in the following.

The first category is called the *confidence-based* approach. The main idea is to use the real-valued output of the classifier as a confidence score [1, 14, 26]. Whether to reject the input is then determined from the classifier's output and a threshold depending on the rejection cost and the choice of the surrogate loss function.

The second category is what we call the *classifier-rejector* approach. Unlike the confidence-based approach, this approach separates the role of the classifier and the rejector, and trains both functions simultaneously [8, 9]. This problem formulation enables more flexible modeling for the rejector, which can be more robust to model-misspecification. This is a state-of-the-art method in binary classification, and has been further discussed in online learning setting [10], structured output learning setting [13], and also in some real-world applications such as liver disease diagnosis [15].

The goal of this paper is to provide a better understanding of multiclass classification with rejection. We first investigate the classifier-rejector approach and derive a calibration condition of this approach in the multiclass case. Our condition recovers the known result by Cortes et al. [9] in the binary case as a special case. However, when there are more than two classes, we argue that the condition is hard to be satisfied. We next analyze the confidence-based approach and prove the calibration results for various classes of smooth losses, which guarantees the use of well-known losses such as the logistic loss, the squared loss, the exponential loss and the cross-entropy loss. Our experiments support the above findings, that is, the failure of the classifier-rejector approach and the success of the confidence-based approach with smooth loss functions, particularly the cross-entropy loss.

## 2 Preliminaries

In this section, we formulate the problem of classification with rejection and review related work.

### 2.1 Problem setting

Let $\mathcal{X} \subseteq \mathbb{R}^d$ be a $d$-dimensional input space and $\mathcal{Y} = \{1, \ldots, K\}$ be an output space representing $K$ classes. Suppose we are given $n$ training samples $\{(\boldsymbol{x}_i, y_i)\}_{i=1}^n$ drawn independently from an unknown probability distribution over $\mathcal{X} \times \mathcal{Y}$ with density $p(\boldsymbol{x}, y)$. In classification with rejection, we will learn a pair $(r, f)$ consisting of a rejector $r$ and a classifier $f$. The rejector $r : \mathcal{X} \to \mathbb{R}$ rejects a point $\boldsymbol{x} \in \mathcal{X}$ if $r(\boldsymbol{x}) \leq 0$, and accepts it otherwise. The classifier $f : \mathcal{X} \to \mathcal{Y}$ is assumed to take the following form:

$$f(\boldsymbol{x}) = \operatorname*{argmax}_{y \in \mathcal{Y}} g_y(\boldsymbol{x}),$$

where $g_y : \mathcal{X} \to \mathbb{R}$ is a score function for multiclass classification. By a slight abuse of notation, we identify the classifier $f(\boldsymbol{x})$ with $\boldsymbol{g}(\boldsymbol{x})$, where $\boldsymbol{g}(\boldsymbol{x}) = [g_1(\boldsymbol{x}), \ldots, g_K(\boldsymbol{x})]^\top$ and $^\top$ denotes the transpose. Given a loss function $\mathcal{L}(r, f; \boldsymbol{x}, y)$, we define its risk $R$ by $R(r, f) = \mathbb{E}_{p(\boldsymbol{x}, y)}[\mathcal{L}(r, f; \boldsymbol{x}, y)]$, where $\mathbb{E}_{p(\boldsymbol{x}, y)}[\cdot]$ denotes the expectation over the distribution $p(\boldsymbol{x}, y)$. We also define the pointwise risk $W$ of the loss $\mathcal{L}$ at $\boldsymbol{x}$ by

$$W\big(r(\boldsymbol{x}), f(\boldsymbol{x}); \boldsymbol{\eta}(\boldsymbol{x})\big) = \sum_{y \in \mathcal{Y}} \eta_y(\boldsymbol{x}) \mathcal{L}\big(r, f; \boldsymbol{x}, y\big),$$

where $\boldsymbol{\eta}(\boldsymbol{x}) = [\eta_1(\boldsymbol{x}), \ldots, \eta_K(\boldsymbol{x})]^\top$ for $\eta_y(\boldsymbol{x}) = p(y|\boldsymbol{x})$ denotes the class probability vector. Note that minimizing $R(r, f)$ with respect to $(r, f)$ over all measurable functions is equivalent to minimizing $W\big(r(\boldsymbol{x}), f(\boldsymbol{x}); \boldsymbol{\eta}(\boldsymbol{x})\big)$ over $\big(r(\boldsymbol{x}), f(\boldsymbol{x})\big)$ for all $\boldsymbol{x} \in \mathcal{X}$. Thus, it is sufficient to only consider the pointwise risk to minimize $R(r, f)$ [22, 25]. For brevity, we omit the notation of $\boldsymbol{x}$ and write, for example, $W(r, f; \boldsymbol{\eta})$ instead of $W\big(r(\boldsymbol{x}), f(\boldsymbol{x}); \boldsymbol{\eta}(\boldsymbol{x})\big)$ for the pointwise risk. We will also drop the notation of $r$ when classification without rejection is considered and write, for example, $\mathcal{L}(f; \boldsymbol{x}, y)$, $R(f)$ and $W(f; \boldsymbol{\eta})$.

In multiclass classification with rejection, our goal is to minimize the 0-1-$c$ risk defined as

$$R_{\text{0-1-}c}(r, f) = \mathbb{E}_{p(\boldsymbol{x}, y)}[\mathcal{L}_{\text{0-1-}c}(r, f; \boldsymbol{x}, y)], \tag{1}$$

where the 0-1-$c$ loss $\mathcal{L}_{\text{0-1-}c}$ is given by

$$\mathcal{L}_{\text{0-1-}c}(r, f; \boldsymbol{x}, y) = \underbrace{\mathbb{1}_{[f(\boldsymbol{x}) \neq y]} \mathbb{1}_{[r(\boldsymbol{x}) > 0]}}_{\text{misclassification loss}} + \underbrace{c \mathbb{1}_{[r(\boldsymbol{x}) \leq 0]}}_{\text{rejection loss}}. \tag{2}$$

Here, $c \in [0, 1]$ denotes the rejection cost, and $\mathbb{1}_{[\cdot]}$ denotes the indicator function.

It is well known that the Bayes-optimal classifier and rejector [20], i.e., the classifier and the rejector that minimize (1), are given by

$$f^*(\boldsymbol{x}) = \operatorname*{argmax}_{y \in \mathcal{Y}} \eta_y(\boldsymbol{x}), \quad r^*(\boldsymbol{x}) = \max_{y \in \mathcal{Y}} \eta_y(\boldsymbol{x}) - (1 - c).$$

In this paper, we assume $c < 1/2$ since data points with low confidence are accepted otherwise.

## 2.2 Calibration

In classification without rejection, the classification risk, i.e., the expected risk with respect to the 0-1 loss $\mathcal{L}_{\text{0-1}}(f; \boldsymbol{x}, y) = \mathbb{1}_{[f(\boldsymbol{x}) \neq y]}$, is the standard performance metric. It is known that minimizing the 0-1 risk is computationally infeasible [4, 12]. Therefore, an important question is what kind of surrogate loss can be used instead of the 0-1 loss [30, 24, 19]. Intuitively, a surrogate loss should be optimization-friendly and its minimization should lead to minimization of the 0-1 risk. The notion of *calibration* is defined for loss functions as the minimum requirement to assure that the risk-minimizing classifier becomes the Bayes-optimal classifier (see Zhang [30] for the formal definition).

In classification with rejection, our goal is to minimize the 0-1-$c$ risk. Similarly to the 0-1 risk, the 0-1-$c$ risk is also difficult to directly minimize [1, 20]. For the purpose of theoretical analysis, it is more convenient to directly define calibration for classifiers and rejectors based on whether they are Bayes-optimal. Thus, we propose to define the notions of calibration as follows.

**Definition 1** (Calibration of a classifier-rejector pair)**.** We say that $(r, f) : \mathcal{X} \to \mathbb{R} \times \mathcal{Y}$ is calibrated if $R_{\text{0-1-}c}(r, f) = R_{\text{0-1-}c}(r^*, f^*)$.

In this paper, we also consider the notions of calibration separately for classifiers and rejectors, which enables better understanding of where the difficulty of classification with rejection comes from.

**Definition 2** (Rejection calibration of a rejector)**.** We say that $r : \mathcal{X} \to \mathbb{R}$ is rejection-calibrated if $\operatorname{sign}[r(\boldsymbol{x})] = \operatorname{sign}[r^*(\boldsymbol{x})]$ for all $\boldsymbol{x} \in \mathcal{X}$ such that $r^*(\boldsymbol{x}) \neq 0$.

**Definition 3** (Classification calibration of a classifier)**.** We say that $f : \mathcal{X} \to \mathcal{Y}$ is classification-calibrated if $f(\boldsymbol{x}) = f^*(\boldsymbol{x})$ holds almost everywhere on $\mathcal{X}$.

As we can see from these definitions and the form of loss function (2), if $r$ is rejection-calibrated and $f$ is classification-calibrated, then $(r, f)$ is calibrated. Furthermore, rejection calibration of $r$ is necessary for calibration of $(r, f)$, while classification calibration of $f$ is not as exemplified in [20].

## 2.3 Related work

Here, we review some related work for both the *confidence-based* and *classifier-rejector* approaches. Note that we follow the conventional notation where the output domain is $\mathcal{Y} = \{+1, -1\}$ and the score function $f : \mathcal{X} \to \mathbb{R}$ is regarded as a classifier when discussing binary classification.

### 2.3.1 Confidence-based approach

In the confidence-based approach, we first train a classifier based on some surrogate of the 0-1 loss, where we regard the real-valued output of the classifier as some confidence score. We then construct a rejector based on the output and a pre-specified threshold $\theta$, which takes the form

$$r(\boldsymbol{x}) = |f(\boldsymbol{x})| - \theta \tag{3}$$

in the binary case. Bartlett and Wegkamp [1] proposed a loss called the modified hinge loss and designed an SVM-like algorithm. Later, Yuan and Wegkamp [29] considered a smooth margin loss $\phi(yf(\boldsymbol{x}))$.

Here the smoothness of the loss is quite important in the construction of rejectors, since the threshold $\theta$ is sometimes not uniquely determined if a non-smooth loss is used. In Bartlett and Wegkamp [1], a calibration guarantee for the non-smooth loss is shown for a range of $\theta$, but its empirical performance is heavily affected by the choice of the threshold. In addition, the loss function also contains a parameter that has to be determined by the rejection cost $c$, which means that we need to re-train the classifier once we change the value of $c$. On the other hand for smooth losses, the value of $c$ does not affect the parameter of a smooth loss, but only the threshold $\theta$. This suggests that we do not need to re-train a classifier when the rejection cost $c$ changes [29].

Ramaswamy et al. [20] extended the method of Bartlett and Wegkamp [1] to multiclass classification, and designed non-smooth losses with excess risk bounds. However, their method has the drawbacks of non-unique $\theta$ and the dependence of the loss on $c$, which comes from the use of non-smooth losses.

### 2.3.2 Classifier-rejector approach

Cortes et al. [8, 9] pointed out that it is too restrictive to require the rejector $r$ to be of form (3) when the true classifier is out of the considered hypothesis set. Based on this observation, they proposed to separate the roles of the classifier and the rejector, and directly minimize an upper bound of the 0-1-$c$ risk with respect to $(r, f)$ in the training phase. Plus bound (PB) loss $\mathcal{L}_{\mathrm{PB}}$ was proposed as an upper bound of the 0-1-$c$ loss in Cortes et al. [9]:

$$\mathcal{L}_{\mathrm{PB}}(r, f; \boldsymbol{x}, y) = \phi\big(\alpha[yf(\boldsymbol{x}) - r(\boldsymbol{x})]\big) + c\psi\big(\beta r(\boldsymbol{x})\big), \tag{4}$$

where $\phi$ and $\psi$ are convex upper bounds of $\mathbb{1}_{[z \leq 0]}$. Cortes et al. [9] derived the calibration result for the exponential loss $\phi(z) = \psi(z) = \exp(-z)$ with appropriately chosen parameters $\alpha, \beta > 0$. However, to the best of our knowledge, this approach is currently available only for the binary case, and an extension to the multiclass case is highly nontrivial as we will see later.

## 3 An analysis of the classifier-rejector approach

In this section, we provide a general result on multiclass classification with rejection using the classifier-rejector approach. In the following, we discuss the achievability of rejection calibration of $r$, which is a necessary condition for calibration of $(r, f)$.

Given a loss $\mathcal{L}(r, f; \boldsymbol{x}, y)$, we denote by $(r_{\boldsymbol{\eta}}^{\dagger}, f_{\boldsymbol{\eta}}^{\dagger}) = \mathrm{argmin}_{r \in \mathbb{R}, \ \boldsymbol{g} \in \mathbb{R}^K} W(r, f; \boldsymbol{\eta})$ the minimizer of the corresponding pointwise risk $W$ over the real space. First we derive the following theorem, which is the main result of this section.

**Theorem 4** (Necessary and sufficient condition for rejection calibration). *Assume that $\mathcal{L}$ is a convex function of class $C^1$ with respect to $r$, and also assume $\left.\frac{\partial^2 W(r, f_{\boldsymbol{\eta}}^{\dagger}; \boldsymbol{\eta})}{\partial r^2}\right|_{r=0} > 0$. Let $(r^{\dagger}, f^{\dagger})$ be the minimizer of the surrogate risk $R$ over all measurable functions. Then, $r^{\dagger}$ is rejection-calibrated if and only if*

$$\sup_{\boldsymbol{\eta}: \max_y \eta_y \geq 1-c} \left.\frac{\partial W(r, f_{\boldsymbol{\eta}}^{\dagger}; \boldsymbol{\eta})}{\partial r}\right|_{r=0} \leq 0 \leq \inf_{\boldsymbol{\eta}: \max_y \eta_y \leq 1-c} \left.\frac{\partial W(r, f_{\boldsymbol{\eta}}^{\dagger}; \boldsymbol{\eta})}{\partial r}\right|_{r=0}. \tag{5}$$

The proof of this theorem is given in Appendix B.1. The following corollary is a weaker version of this theorem but gives more insight into the strength of the requirement for rejection calibration.

**Corollary 5** (Necessary condition for rejection calibration). *Under the same assumption as Theorem 4, $r^{\dagger}$ is rejection-calibrated only if*

$$\sup_{\boldsymbol{\eta}: \max_y \eta_y = 1-c} \left.\frac{\partial W(r, f_{\boldsymbol{\eta}}^{\dagger}; \boldsymbol{\eta})}{\partial r}\right|_{r=0} = 0, \qquad \inf_{\boldsymbol{\eta}: \max_y \eta_y = 1-c} \left.\frac{\partial W(r, f_{\boldsymbol{\eta}}^{\dagger}; \boldsymbol{\eta})}{\partial r}\right|_{r=0} = 0. \tag{6}$$

This corollary is straightforward from the relation

$$\inf_{\boldsymbol{\eta}: \max_y \eta_y \leq 1-c} h(\boldsymbol{\eta}) \leq \inf_{\boldsymbol{\eta}: \max_y \eta_y = 1-c} h(\boldsymbol{\eta}) \leq \sup_{\boldsymbol{\eta}: \max_y \eta_y = 1-c} h(\boldsymbol{\eta}) \leq \sup_{\boldsymbol{\eta}: \max_y \eta_y \geq 1-c} h(\boldsymbol{\eta})$$

for any function $h(\boldsymbol{\eta})$. The conditions in (6) require that the supremum and the infimum of the objective function $\left.\frac{\partial W(r, f_{\boldsymbol{\eta}}^{\dagger}; \boldsymbol{\eta})}{\partial r}\right|_{r=0}$ coincide under the same constraint. Therefore, the objective function is required to depend only on $\max_y \eta_y$, but not on the class probabilities of other classes. Whereas $\max_y \eta_y$ uniquely determines the other probability as $1 - \max_y \eta_y$ in the binary case, it still allows a degree of freedom in the multiclass case, which results in the situation where two conditions in (6) do not necessarily hold simultaneously.

The failure of the classifier-rejector approach is intuitively explained as follows. The Bayes-optimal rejector $r^*$ must be determined only from $\max_y \eta_y$. Nevertheless, the classifier-rejector approach

ignores this requirement and tries to directly construct a rejector $r$, which does not satisfy this requirement in general. This contrasts to the rejector in (10) obtained by the confidence-based approach, where the requirement is encoded by the inverse link function and the max operator.

**Remark 1.** An error of the rejector can be classified into *False Reject* (FR) and *False Accept* (FA), which correspond to the outcomes when the rejector mistakenly rejects (resp. accepts) the data that should be accepted (resp. rejected). We can see from close inspection of the proof of Theorem 4 that the first inequality of (5) is the condition for the FR rate to be zero, while the second inequality is the condition for the FA rate to be zero.

To understand the above difference between the binary and multiclass cases more precisely, let us consider the following example so that the conditions in (6) are explicitly written. Define two surrogate losses given by

$$\mathcal{L}_{\mathrm{MPC}}(r, f; \boldsymbol{x}, y) = \sum_{y' \neq y} \phi\Big(\alpha\big(g_y(\boldsymbol{x}) - g_{y'}(\boldsymbol{x})\big)\Big)\psi(-\alpha r(\boldsymbol{x})) + c\psi\big(\beta r(\boldsymbol{x})\big), \qquad (7)$$

$$\mathcal{L}_{\mathrm{APC}}(r, f; \boldsymbol{x}, y) = \sum_{y' \neq y} \phi\Big(\alpha\big(g_y(\boldsymbol{x}) - g_{y'}(\boldsymbol{x}) - r(\boldsymbol{x})\big)\Big) + c\psi\big(\beta r(\boldsymbol{x})\big), \qquad (8)$$

which we call the multiplicative pairwise comparison (MPC) loss and the additive pairwise comparison (APC) loss, respectively. Here, $\phi$ and $\psi$ are convex losses that bound $\mathbb{1}_{[z \leq 0]}$ from above, and $\alpha$ and $\beta$ are positive constants that control the performance of the rejector. Note that the pairwise comparison loss is often used as a multiclass extension of a binary loss [27]. Also note that the APC loss reduces to the PB loss [9] in (4) when $K = 2$. Here the MPC loss and the APC loss are natural ones at least for the purpose of classification in the sense that the classifiers induced by them are classification-calibrated (see Appendix B.3 for the proof). Nevertheless, when $\phi$ and $\psi$ are exponential losses, (6) gives the following conditions:

$$\frac{\beta}{\alpha} = (K - 2) + 2\sqrt{(K - 1)\frac{1 - c}{c}}, \qquad \frac{\beta}{\alpha} = 2\sqrt{\frac{1 - c}{c}}, \qquad (9)$$

which recover the result proved by Cortes et al. [9] when $K = 2$ (see Appendix B.4 for details). Here the RHSs of (9) for $K > 2$ are not identical and therefore we cannot find any $\alpha$ and $\beta$ satisfying the above equations, even though we get a classification-calibrated classifier. This implies the failure in rejection calibration. Not only when $\phi$ and $\psi$ are exponential losses, we can also prove the failure of the classifier-rejector approach when $\phi$ and $\psi$ are logistic losses using the same proof technique (see Appendix B.4).

Note that, strictly speaking, it remains an open question whether it is possible to find a calibrated surrogate loss in the classifier-rejector approach. In this paper, our result emphasizes that calibration in the multiclass scenario is significantly more difficult. Intuitively, a necessary condition in Corollary 5 is relatively easy to satisfy for $K = 2$ but it is not the case when $K > 2$, as illustrated in our examples.

## 4 An analysis of the confidence-based approach

This section focuses on the extension of the confidence-based approach to the multiclass case using smooth losses. When we need some confidence score in the multiclass case, it is convenient to consider a class of loss functions called strictly proper composite losses [22] defined as follows.

**Definition 6** (Strictly proper composite loss [22]). A loss $\mathcal{L}$ is strictly proper composite with link function $\boldsymbol{\Psi} : [0, 1]^K \to \mathbb{R}^K$ if the pointwise risk $W$ of $\mathcal{L}$ satisfies $\arg\min_{\boldsymbol{g}} W(\boldsymbol{g}; \boldsymbol{\eta}) = \boldsymbol{\Psi}(\boldsymbol{\eta}) = [\Psi_1(\boldsymbol{\eta}), \ldots, \Psi_K(\boldsymbol{\eta})]^\top$.

With this class of losses, the threshold $\theta$ used in the rejector derived in Yuan and Wegkamp [29] is expressed as $\Psi_1\big((1 - c, c)\big)$ in the binary case. However, unlike the binary case, it is known that the link function $\boldsymbol{\Psi}$ sometimes does not have a closed form whereas the inverse link function $\boldsymbol{\Psi}^{-1}$ often does in multiclass classification [22]. Thus, when we design a rejector in the multiclass case, it would be natural to use the inverse link function to map output $\widehat{\boldsymbol{g}}$ to the estimated class probability vector $\widehat{\boldsymbol{\eta}}$ rather than to use the link function itself as in the binary case. Based on this discussion, we consider the following rejector based on the relationship between the inverse link $\Psi_y^{-1}$ and the Bayes-optimal

Table 1: A list of margin losses and the values of $\theta$, $C$ and $s$ that satisfy (14) and (15) in Theorem 7.

| Loss Name | $\phi(z)$ | $\theta$ | $C$ | $s$ |
|---|---|---|---|---|
| Logistic | $\log\big(1 + \exp(-z)\big)$ | $\log\frac{1-c}{c}$ | $\frac{1}{2}$ | 2 |
| Exponential | $\exp(-z)$ | $\frac{1}{2}\log\frac{1-c}{c}$ | $\frac{1}{\sqrt{2}}$ | 2 |
| Squared | $(1-z)^2$ | $1-2c$ | $\frac{1}{2}$ | 2 |
| Squared Hinge | $(1-z)_+^2$ | $1-2c$ | $\frac{1}{2}$ | 2 |

rejector $r^*(\boldsymbol{x}) = \max_{y \in \mathcal{Y}} \eta_y(\boldsymbol{x}) - (1-c)$:

$$r(\boldsymbol{x}) = r_f(\boldsymbol{x}) = \max_{y \in \mathcal{Y}} \Psi_y^{-1}\big(\boldsymbol{g}(\boldsymbol{x})\big) - (1-c). \tag{10}$$

Recall that we identify the classifier $f$ with $\boldsymbol{g}$, and we use the notation $r_f$ in the sense that $r$ is determined by $f$. Below, we focus on two frequently used losses: one-versus-all (OVA) loss $\mathcal{L}_{\text{OVA}}$ and cross-entropy (CE) loss $\mathcal{L}_{\text{CE}}$:

$$\mathcal{L}_{\text{OVA}}(f; \boldsymbol{x}, y) = \phi\big(g_y(\boldsymbol{x})\big) + \sum_{y' \neq y} \phi\big(-g_{y'}(\boldsymbol{x})\big), \tag{11}$$

$$\mathcal{L}_{\text{CE}}(f; \boldsymbol{x}, y) = -g_y(\boldsymbol{x}) + \log \sum_{y' \in \mathcal{Y}} \exp\big(g_{y'}(\boldsymbol{x})\big),$$

for which the inverse link functions are given by

$$\Psi_{y, \text{OVA}}^{-1}(\boldsymbol{g}) = \frac{\phi'(-g_y)}{\phi'(-g_y) + \phi'(g_y)}, \qquad \Psi_{y, \text{CE}}^{-1}(\boldsymbol{g}) = \frac{\exp(g_y)}{\sum_{y' \in \mathcal{Y}} \exp(g_{y'})}, \tag{12}$$

respectively. Here, $\phi$ denotes a margin loss [3]. Note that unlike the losses proposed in Ramaswamy et al. [20], the OVA loss and the CE loss do not contain $c$. Thus, training a classifier once is sufficient for various choices of $c$.

We rely on the notion of excess risk bounds to prove the calibration result of the OVA loss and the CE loss. Excess risk bounds [30, 3, 19] are a tool to directly quantify the relationship between the surrogate risk $R$ and the risk we truly want to minimize. In our problem, the true risk is the 0-1-$c$ risk in (1) and the excess risk bound to be derived is expressed as

$$\xi\big(\Delta R_{\text{0-1-}c}(r_f, f)\big) \leq \Delta R(f), \tag{13}$$

where $\xi : \mathbb{R} \to \mathbb{R}_{\geq 0}$ is called a calibration function [19], which is increasing, continuous at 0 and satisfies $\xi(0) = 0$. Here, excess risks $\Delta R_{\text{0-1-}c}(r_f, f)$ and $\Delta R(f)$ are defined as follows:

$$\Delta R_{\text{0-1-}c}(r_f, f) = R_{\text{0-1-}c}(r_f, f) - R_{\text{0-1-}c}(r^*, f^*),$$
$$\Delta R(f) = R(f) - \inf_{f': \text{measurable}} R(f').$$

Ineq. (13) ensures that the minimization of a surrogate risk leads to the minimization of the 0-1-$c$ risk. Therefore, the existence of an excess risk bound guarantees calibration.

Now we give excess risk bounds for the OVA loss and the CE loss in the following theorems.

**Theorem 7** (Excess risk bound for OVA loss). *Assume that $\phi$ is a convex function, and there exists $\theta > 0$ such that $\phi'(\theta)$ and $\phi'(-\theta)$ both exist, $\phi'(\theta) < 0$ and*

$$\frac{\phi'(-\theta)}{\phi'(-\theta) + \phi'(\theta)} = 1 - c. \tag{14}$$

*In addition, suppose that there exist some constants $C > 0$ and $s \geq 1$ such that*

$$\inf_{\boldsymbol{g}:\, g_y = \theta} \left\{ W_{\text{OVA}}(f; \boldsymbol{\eta}) - \inf_{\boldsymbol{g}' \in \mathbb{R}^K} W_{\text{OVA}}(f'; \boldsymbol{\eta}) \right\} \geq C^{-s} |\eta_y - (1-c)|^s \tag{15}$$

*for all $y \in \mathcal{Y}$ and probability vector $\boldsymbol{\eta}$. Then, for all $f$ and $c \in \big[0, \frac{1}{2}\big)$, we have*

$$(2C)^{-s} \Delta R_{\text{0-1-}c}(r_f, f)^s \leq \Delta R_{\text{OVA}}(f). \tag{16}$$

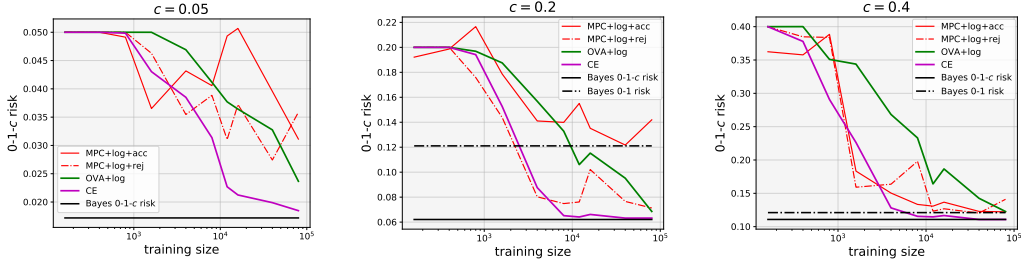

Figure 1: Average 0-1-$c$ risk on the test data as a function of the training data size on synthetic datasets.

Table 1 summarizes some margin losses with the values of $\theta$, $C$ and $s$ that satisfy the assumptions (14) and (15). Their derivations are given in Appendix A.2.

**Theorem 8** (Excess risk bound for CE loss). *For all $f$ and $c \in (0, 1/2)$, we have*

$$\frac{1}{2}\Delta R_{0\text{-}1\text{-}c}(r_f, f)^2 \leq \Delta R_{\text{CE}}(f).$$

The proofs of Theorems 7 and 8 can be found in Appendices A.1 and A.4, respectively. The derivation of the bound for the OVA loss is a natural extension of Yuan and Wegkamp [29] for the binary case. On the other hand, although the CE loss can be regarded as a generalization of the logistic loss in binary classification, the derivation of the excess risk bound for the logistic loss in Yuan and Wegkamp [29] heavily relies on the binary setting and is not applicable to the multiclass case. In fact, the CE loss is generally hard to bound even in the setting without rejection as discussed in Pires and Szepesvári [19]. In this paper, we reduce the analysis of the CE loss into that of the KL divergence instead of trying to extend the argument of Yuan and Wegkamp [29] or Pires and Szepesvári [19]. This enabled us to derive the bound in a considerably simple way.

The excess risk bounds in Theorems 7 and 8 ensure that the minimization of the expected surrogate risk leads to the minimization of the 0-1-$c$ risk. On the other hand, we can also derive an estimation error bound for the above losses, which shows that the minimization of the empirical surrogate risk leads to the minimization of the expected surrogate risk for a finite number of samples with a hypothesis class of our interest. Combining these results completes the scenario to minimize the 0-1-$c$ risk from finite number of samples under the considered hypothesis class. Here the derivation of the estimation error bound using the notion of Rademacher complexity [2] is given in Appendix A.3.

## 5 Experiments

In this section, we report the results of two experiments based on synthetic and benchmark datasets. The purpose of the experiment on synthetic datasets is to verify the performance of calibration for the setting where Bayes-optimal 0-1-c risk is available. On the other hand, we use benchmark datasets to evaluate the practical performance.

**Common setup:** For all methods, we used one-hidden-layer neural networks with the rectified linear units (ReLU) as activation functions, where the number of hidden units is 3 for synthetic datasets, and 50 for benchmark datasets. We added weight decay with candidates $\{10^{-7}, 10^{-4}, 10^{-1}\}$. AMSGRAD [21] was used for optimization. More detailed setups can be found in Appendix C.

**Synthetic datasets:** Here we report the performance of four methods analyzed in this paper. For the classifier-rejector approach, we used the MPC loss with the logistic loss in (7), where we used $\alpha = 1$ as in Cortes et al. [9]. To see the performance of the rejector, we set two values for $\beta$ to satisfy either of (6) denoted by MPC+log+acc and MPC+log+rej, respectively. It is expected that MPC+log+acc will over-accept the data, and MPC+log+rej will over-reject the data as discussed in Remark 1. For the confidence-based approach, we used the CE loss (CE) and OVA loss with the logistic loss in (11) denoted by OVA+log. Synthetic data consist of eight classes. More detailed information on data generation process can be found in Appendix C.1.

Figure 1 shows the average 0-1-$c$ risk on the test data for various training data size, where the lower 0-1-$c$ risk is the better. CE shows the best performance in terms of convergence to the Bayes-optimal

Table 2: Mean and standard deviation of the ratio (%) of the rejected data over all test data on synthetic datasets when the training data size is 10,000 per class.

| $c$ | MPC+log+acc | MPC+log+rej | OVA+log | CE |
|---|---|---|---|---|
| 0.05 | 25.4 (8.6) | 46.4 (7.6) | 43.9 (1.3) | 33.9 (0.5) |
| 0.2 | 0.0 (0.0) | 23.2 (1.6) | 31.5 (0.3) | 28.3 (1.5) |
| 0.4 | 0.0 (0.0) | 28.8 (9.9) | 23.1 (0.7) | 17.3 (0.8) |

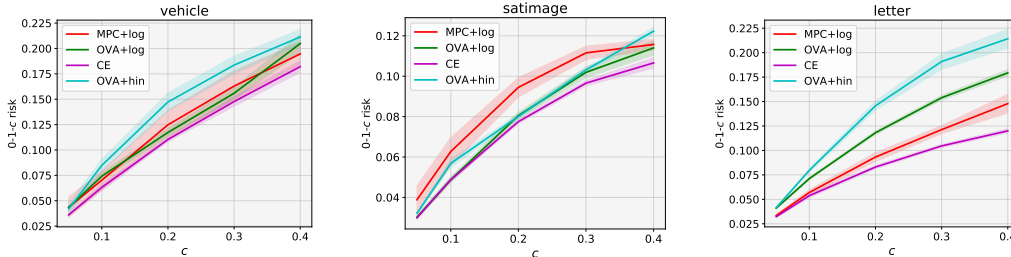

Figure 2: Average and standard error of $0\text{-}1\text{-}c$ risk on the test data as a function of rejection cost $c$ on benchmark datasets for 10 trials. The standard error is plotted in shaded regions.

$0\text{-}1\text{-}c$ risk for all values of $c$. In spite of the theoretical guarantees of the confidence-based methods of OVA losses, they did not show better performance than the others. A possible reason is that the inverse link function of the OVA loss is not normalized as can be seen from (12), which resulted in poor estimation of class probability $\boldsymbol{\eta}(\boldsymbol{x})$. It is observed that the classifier-rejector methods (MPC+log+acc and MPC+log+rej) show unstable performance compared to the other methods. Table 2 shows the rejection ratio when the training data size is 10,000 per class. We can confirm that MPC+log+acc tends to over-accept and MPC+log+rej tends to over-reject the data, which agrees with the discussion in Remark 1.

**Benchmark datasets:** We compared the empirical performance using benchmark datasets with rejection cost ranged over $c \in \{0.05, 0.1, 0.2, 0.3, 0.4\}$. In addition to APC+log, MPC+log, OVA+log and CE, we further implemented the existing method proposed in Ramaswamy et al. [20] (OVA+hin), which uses OVA loss with non-smooth hinge loss in (11). We show the results of *vehicle*, *satimage*, *yeast*, *covtype* and *letter* datasets from *UCI Machine Learning Repository* [17], which are the same datasets as those used in Ramaswamy et al. [20]. Table 3 summarizes the specification of the benchmark datasets we used. For the classifier-rejector methods (APC+log, MPC+log), we have extra parameters $\alpha$ and $\beta$. We set $\alpha = 1$ as in Cortes et al. [9]. We chose $\beta$ by cross-validation, where the choices of $\beta$ that satisfies either of (6) were also included. In the OVA+hin formulation, Ramaswamy et al. [20] suggested that the threshold parameter $\tau \in (-1, 1)$ in their methods is preferable at 0. Nevertheless, we observed that the performance is considerably affected by its choice and thus we decided to choose the best parameter from five candidates by cross-validation. See Appendix C.2 for the detailed information on experimental setups. Note that APC+log, MPC+log and OVA+hin must be re-trained for different rejection costs, while OVA+log and CE do not need re-training. The full experimental results including the performance of other methods, the full report of the $0\text{-}1\text{-}c$ risk, the accuracy of the non-rejected data, and the rejection ratio can be found in Appendix C.

Figure 2 illustrates the $0\text{-}1\text{-}c$ risk as functions of the rejection cost. It can be observed that CE is either competitive or preferable in all datasets. For OVA+log, despite its calibration guarantees, it is outperformed by CE for all datasets and it is even outperformed by MPC+log in *letter* dataset. The failure of the OVA methods in *letter* might be due to their weakness for a large number of classes [5] and poor estimation of $\boldsymbol{\eta}(\boldsymbol{x})$. It is also worth noting that the standard deviations of MPC+log and OVA+hin are considerably large compared to those of OVA+log and CE, which might be caused by additional hyper-parameters $\beta$ and $\tau$. Moreover, model fitting for a rejector and the non-convexity of the MPC loss function also make MPC+log unstable. Table 4 shows the mean and standard deviation of the accuracy on non-rejected data. As we can see clearly in *yeast* datasets, unlike the confidence-based methods, the classifier-rejector methods reject all the test data even when the value

Table 3: Specification of benchmark datasets: the number of features, the number of classes, the number of training data, and the number of test data.

| Name | #features | #classes | #train | #test |
|---|---|---|---|---|
| vehicle | 18 | 4 | 700 | 146 |
| satimage | 36 | 6 | 4435 | 2000 |
| yeast | 8 | 10 | 1000 | 484 |
| covtype | 54 | 7 | 15120 | 565892 |
| letter | 16 | 26 | 15000 | 5000 |

Table 4: Mean and standard deviation of the accuracy (%) of the non-rejected data samples for 10 trials. Best and equivalent methods (with 5% t-test) with respect to the $0$-$1$-$c$ risk are shown in bold face. "–" corresponds to the case where all the test data samples are rejected.

| dataset | $c$ | APC+log | MPC+log | OVA+log | CE |
|---|---|---|---|---|---|
| vehicle | 0.05 | – ( – ) | 96.6 (2.3) | 100 (0.0) | **100 (0.0)** |
| | 0.2 | 98.4 (1.9) | 92.4 (3.0) | 97.9 (0.7) | **97.4 (0.1)** |
| | 0.4 | **89.1 (2.9)** | 85.3 (4.2) | 90.2 (1.6) | **91.7 (0.9)** |
| satimage | 0.05 | **99.1 (0.2)** | 97.2 (1.4) | **98.7 (0.1)** | 98.3 (0.1) |
| | 0.2 | 95.0 (1.0) | 92.6 (1.2) | 96.2 (0.2) | **95.7 (0.1)** |
| | 0.4 | 91.5 (0.7) | 89.0 (1.1) | 92.2 (0.3) | **91.8 (0.2)** |
| yeast | 0.05 | – ( – ) | – ( – ) | – ( – ) | – ( – ) |
| | 0.2 | – ( – ) | – ( – ) | – ( – ) | **80.6 (6.2)** |
| | 0.4 | – ( – ) | – ( – ) | 75.0 (3.9) | **76.6 (1.7)** |

| dataset | $c$ | APC+log | MPC+log | OVA+log | CE |
|---|---|---|---|---|---|
| covtype | 0.05 | **79.5 (2.1)** | 79.8 (1.7) | **82.1 (2.7)** | 82.0 (3.2) |
| | 0.2 | 74.0 (1.8) | 73.8 (1.0) | 74.9 (1.4) | **77.1 (0.3)** |
| | 0.4 | **69.8 (1.3)** | 64.9 (3.4) | **68.7 (1.1)** | 69.4 (1.8) |
| letter | 0.05 | **99.8 (0.1)** | 98.6 (0.2) | **99.6 ( 0.2 )** | 99 8 (0.0) |
| | 0.2 | 97.9 (0.3) | 96.9 (0.5) | **98.3 (0.2)** | 98.4 (0.1) |
| | 0.4 | **95.2 (0.5)** | 94.6 (3.8) | 94.6 (0.2) | **94.9 (0.3)** |

of $c$ is large. This implies that if the dataset is hard to learn, then classifier-rejector methods may fail to learn the rejector.

# 6    Conclusion

We presented a series of theoretical results on multiclass classification with rejection. First, we provided a necessary condition of rejection calibration for the classifier-rejector approach that suggested the difficulty of calibration for this approach in the multiclass case. Second, we investigated the confidence-based approach and established the calibration results for the OVA loss and the CE loss by deriving excess risk bounds. Experimental results suggested that the CE loss is the most preferable and the classifier-rejector approach can no longer outperform the confidence-based methods unlike the binary case.

## Acknowledgements

We thank Han Bao for fruitful discussions. We also thank anonymous reviewers for providing insightful comments. NC was supported by MEXT scholarship and JST AIP challenge. JH was supported by KAKENHI 18K17998. MS was supported by the International Research Center for Neurointelligence (WPI-IRCN) at The University of Tokyo Institutes for Advanced Study.

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
