[Supplementary Material]

# A Proofs

Table 5: Cases we discuss in the proofs of Theorems 7 and 8.

| $\Psi_f$ | $\Psi_f^{-1}(\boldsymbol{g}) > 1 - c$ | | $\Psi_f^{-1}(\boldsymbol{g}) \leq 1 - c$ | |
|---|---|---|---|---|
| $\eta_{y^*}$ | $f = y^*$ | $f \neq y^*$ | $f = y^*$ | $f \neq y^*$ |
| $\eta_{y^*} > 1 - c$ | (A) | (B) | (C) | (D) |
| $\eta_{y^*} \leq 1 - c$ | (E) | (F) | (G) | (H) |

To begin with, define the excess pointwise 0-1-$c$ risk as

$$\Delta W_{\text{0-1-}c}(r, f; \boldsymbol{\eta}) = W_{\text{0-1-}c}(r, f; \boldsymbol{\eta}) - \min \left\{ c, 1 - \max_{y \in \mathcal{Y}} \eta_y \right\}. \tag{17}$$

## A.1 Proof of Theorem 7

The pointwise risk of OVA loss is expressed as

$$W_{\text{OVA}}(\boldsymbol{g}; \boldsymbol{\eta}) = \sum_y \left[ \eta_y \phi(g_y) + (1 - \eta_y)\phi(-g_y) \right]. \tag{18}$$

We write the the excess pointwise risk of OVA loss as

$$\Delta W_{\text{OVA}}(f; \boldsymbol{\eta}) = W_{\text{OVA}}(f; \boldsymbol{\eta}) - \inf_{\boldsymbol{g} \in \mathbb{R}^K} W_{\text{OVA}}(f; \boldsymbol{\eta}).$$

The main focus of this proof is to show the following inequality:

$$\Delta W_{\text{0-1-}c}(r, f; \boldsymbol{\eta}) \leq 2C\Delta W_{\text{OVA}}(\boldsymbol{g}; \boldsymbol{\eta})^{\frac{1}{s}}, \tag{19}$$

since if the above inequality holds, then Ineq. (16) can be derived as follows:

$$\begin{aligned}
\Delta R_{\text{0-1-}c}(r_f, f) &= \mathop{\mathbb{E}}_{\boldsymbol{x} \sim p(\boldsymbol{x})} \left[ \Delta W_{\text{0-1-}c}\big(r_f(\boldsymbol{x}), f(\boldsymbol{x}); \boldsymbol{\eta}(\boldsymbol{x})\big) \right] \\
&\leq \mathop{\mathbb{E}}_{\boldsymbol{x} \sim p(\boldsymbol{x})} \left[ 2C\Delta W_{\text{OVA}}\big(\boldsymbol{g}(\boldsymbol{x}); \boldsymbol{\eta}(\boldsymbol{x})\big)^{\frac{1}{s}} \right] \\
&\leq 2C \mathop{\mathbb{E}}_{\boldsymbol{x} \sim p(\boldsymbol{x})} \left[ \Delta W_{\text{OVA}}\big(\boldsymbol{g}(\boldsymbol{x}); \boldsymbol{\eta}(\boldsymbol{x})\big) \right]^{\frac{1}{s}} \\
&\leq 2C R_{\text{OVA}}(\boldsymbol{g})^{\frac{1}{s}},
\end{aligned} \tag{20}$$

where we used Jensen's inequality in (20). To prove Ineq. (19), we need to consider the different cases with respect to $\boldsymbol{\eta}$ and $\boldsymbol{\Psi}(\boldsymbol{g})$, which is summarized in Table 5. Again, we will abbreviate $\boldsymbol{x}$ for brevity in the rest of this proof. Note that $\Psi_f(\boldsymbol{x}) = \max_{y \in \mathcal{Y}} \Psi_y(\boldsymbol{x})$ if we use proper composite loss.

**Cases (A)(G)(H):**
In this case, we can confirm that $\Delta W_{\text{0-1-}c}(r_f, f; \boldsymbol{\eta}) = 0$ by (17) since $(r_f, f)$ makes a correct prediction. Thus, it holds that

$$\Delta W_{\text{0-1-}c}(r_f, f; \boldsymbol{\eta}) = 0 \leq 2C\Delta W_{\text{OVA}}(\boldsymbol{g}; \boldsymbol{\eta})^{\frac{1}{s}}.$$

**Cases (C)(D):**
In this case, we can confirm that

$$\Delta W_{\text{0-1-}c}(r_f, f; \boldsymbol{\eta}) = c - (1 - \eta_{y^*}) \tag{21}$$

by (17). Let $\boldsymbol{g}^\dagger = \operatorname{argmin}_{g'_{y^*} = \theta} W_{\text{OVA}}(\boldsymbol{g}'; \boldsymbol{\eta})$. Using the Lagrangian multiplier method we see that $g^\dagger$ has to satisfy

$$\frac{\phi'(-g_y^\dagger)}{\phi'(-g_y^\dagger) + \phi'(g_y^\dagger)} = \begin{cases} \eta_y & (y \neq y^*), \\ 1 - c & (y = y^*), \end{cases} \tag{22}$$

and the LHS of which actually exists by the assumption (14). We now prove

$$W_{\mathrm{OVA}}(\boldsymbol{g};\boldsymbol{\eta}) - W_{\mathrm{OVA}}(\boldsymbol{g}^\dagger;\boldsymbol{\eta}) \geq 0. \tag{23}$$

Recall that $W_{\mathrm{OVA}}$ (18) is a convex combination of $\phi$, which is a convex function. Thus, $W_{\mathrm{OVA}}$ is also a convex function. Therefore,

$$
\begin{aligned}
W_{\mathrm{OVA}}(\boldsymbol{g};\boldsymbol{\eta}) - W_{\mathrm{OVA}}(\boldsymbol{g}^\dagger;\boldsymbol{\eta}) &\geq \nabla_{\boldsymbol{g}}\, W_{\mathrm{OVA}}(\boldsymbol{g};\boldsymbol{\eta})\big|_{\boldsymbol{g}=\boldsymbol{g}^\dagger}^\top (\boldsymbol{g}-\boldsymbol{g}^\dagger) \\
&= \sum_{y\neq y^*} \underbrace{\left[\eta_y\phi'(g_y^\dagger) - (1-\eta_y)\phi'(-g_y^\dagger)\right]}_{=0\ (\text{using } (22))}(g_y - g_y^\dagger) \\
&\quad + \left[\eta_{y^*}\phi'(g_{y^*}^\dagger) - (1-\eta_{y^*})\phi'(-g_{y^*}^\dagger)\right](g_{y^*} - g_{y^*}^\dagger) \\
&= \left[\eta_{y^*}\phi'(g_{y^*}^\dagger) - (1-\eta_{y^*})\phi'(-g_{y^*}^\dagger)\right](g_{y^*} - g_{y^*}^\dagger).
\end{aligned}
$$

The condition $\eta_{y^*} > 1 - c$ together with the assumption $\phi'(-g_{y^*}^\dagger) \leq \phi'(g_{y^*}^\dagger) = \phi'(\theta) < 0$ gives

$$\eta_{y^*}\phi'(g_{y^*}^\dagger) - (1-\eta_{y^*})\phi'(-g_{y^*}^\dagger) < (1-c)\phi'(g_{y^*}^\dagger) - c\phi'(-g_{y^*}^\dagger) = 0.$$

Here, we used (22) in the last equality. In addition, since we have $\Psi_{y^*}^{-1}(\boldsymbol{g}) \leq \Psi_f^{-1}(\boldsymbol{g}) \leq 1-c$ in cases (C)(D), together with $\Psi_{y^*}^{-1}(\boldsymbol{g}^\dagger) = 1 - c$, we get the inequality $\Psi_{y^*}^{-1}(\boldsymbol{g}) < \Psi_{y^*}^{-1}(\boldsymbol{g}^\dagger)$. Note that $\Psi_{y^*}^{-1}$ is a non-decreasing function with respect to $g_{y^*}$, so it holds that

$$g_{y^*} \leq g_{y^*}^\dagger.$$

Therefore, we can conclude that (23) holds, which gives the following result:

$$
\begin{aligned}
\Delta W_{\mathrm{OVA}}(\boldsymbol{g};\boldsymbol{\eta}) &\geq \inf_{g'_{y^*}=\theta} \Delta W_{\mathrm{OVA}}(\boldsymbol{g}';\boldsymbol{\eta}) \\
&\geq C^{-s}|\eta_{y^*} - (1-c)|^s &&\text{(by assumption (15))} \\
&= C^{-s}\Delta W_{\text{0-1-}c}(r_f, f;\boldsymbol{\eta})^s. &&\text{(by using (21))}
\end{aligned}
$$

**Cases (E)(F):**

In this case, we can confirm that

$$\Delta W_{\text{0-1-}c}(r_f, f;\boldsymbol{\eta}) = |(1-\eta_f) - c| \tag{24}$$

by (17). Let $\boldsymbol{g}^\sharp = \operatorname{argmin}_{g'_f=\theta} W_{\mathrm{OVA}}(\boldsymbol{g}';\boldsymbol{\eta})$. Similarly to the above case, the optimal solution $\boldsymbol{g}^\sharp$ satisfies the following:

$$\frac{\phi'(-g_y^\sharp)}{\phi'(-g_y^\sharp) + \phi'(g_y^\sharp)} = \begin{cases} \eta_y & (y \neq f), \\ 1-c & (y = f), \end{cases} \tag{25}$$

and the LHS of which actually exists by the assumption (14). We now prove

$$W_{\mathrm{OVA}}(\boldsymbol{g};\boldsymbol{\eta}) - W_{\mathrm{OVA}}(\boldsymbol{g}^\sharp;\boldsymbol{\eta}) \geq 0. \tag{26}$$

Recall that $W_{\mathrm{OVA}}$ (18) is a convex combination of $\phi$, which is a convex function. Thus, $W_{\mathrm{OVA}}$ is also a convex function. Therefore,

$$
\begin{aligned}
W_{\mathrm{OVA}}(\boldsymbol{g};\boldsymbol{\eta}) - W_{\mathrm{OVA}}(\boldsymbol{g}^\sharp;\boldsymbol{\eta}) &\geq \nabla_{\boldsymbol{g}}W_{\mathrm{OVA}}(\boldsymbol{g}^\sharp;\boldsymbol{\eta})^\top (\boldsymbol{g}-\boldsymbol{g}^\sharp) \\
&= \sum_{y\neq f} \underbrace{\left[\eta_y\phi'(g_y^\sharp) - (1-\eta_y)\phi'(-g_y^\sharp)\right]}_{=0\ (\text{using } (25))}(g_y - g_y^\sharp) \\
&\quad + \left[\eta_f\phi'(g_f^\sharp) - (1-\eta_f)\phi'(-g_f^\sharp)\right](g_f - g_f^\sharp) \\
&= \left[\eta_f\phi'(g_f^\sharp) - (1-\eta_f)\phi'(-g_f^\sharp)\right](g_f - g_f^\sharp).
\end{aligned}
$$

The condition $\eta_f \le \eta_{y^*} \le 1 - c$ together with the assumption $\phi'(-g_f^\sharp) \le \phi'(-g_f^\sharp) = \phi'(\theta) < 0$ gives

$$\eta_f \phi'(g_f^\dagger) - (1 - \eta_f)\phi'(-g_f^\sharp) \ge (1 - c)\phi'(g_f^\dagger) - c\phi'(-g_f^\sharp) = 0.$$

Here, we used (25) in the last equality. In addition, since we have $\Psi_f^{-1}(\boldsymbol{g}) > 1 - c$ in cases (E)(F), together with $\Psi_f^{-1}(\boldsymbol{g}^\sharp) = 1 - c$, we get the inequality $\Psi_f^{-1}(\boldsymbol{g}) > \Psi_f^{-1}(\boldsymbol{g}^\sharp)$. Note that $\Psi_f^{-1}$ is a non-decreasing function with respect to $g_f$, so it holds that

$$g_f \ge g_f^\sharp.$$

Therefore, we can conclude that (26) holds, which gives the following result:

$$\Delta W_{\mathrm{OVA}}(\boldsymbol{g}; \boldsymbol{\eta}) \ge \inf_{g_f' = \theta} \Delta W_{\mathrm{OVA}}(\boldsymbol{g}'; \boldsymbol{\eta})$$

$$\ge C^{-s}|\eta_f - (1 - c)|^s \qquad \text{(by assumption (15))}$$

$$\ge C^{-s}\Delta W_{\text{0-1-}c}(r_f, f; \boldsymbol{\eta})^s. \qquad \text{(by using (24))}$$

**Case (B):**

In this case, we can confirm that

$$\Delta W_{\text{0-1-}c}(r_f, f; \boldsymbol{\eta}) = \eta_{y^*} - \eta_f \qquad (27)$$

by (17). Again, we will prove (26) and utilize the property (25) of optimal solution $\boldsymbol{g}^\sharp$. By assumption $c < \frac{1}{2}$ and property $\sum_y \eta_y = 1$, we have $\eta_{y^*} > 1 - c > \eta_f$. This inequality together with the assumption $\phi'(-g_f^\sharp) \le \phi'(g_f^\sharp) = \phi'(\theta) < 0$ gives

$$\eta_f \phi'(g_f^\sharp) - (1 - \eta_f)\phi'(-g_f^\sharp) > (1 - c)\phi'(g_f^\sharp) - c\phi'(-g_f^\sharp) = 0.$$

Here, we used (25) in the last equality. In addition, since we have $\Psi_f^{-1}(\boldsymbol{g}) > 1 - c$ in case (B), together with $\Psi_f^{-1}(\boldsymbol{g}^\sharp) = 1 - c$, we get the inequality $\Psi_f^{-1}(\boldsymbol{g}) > \Psi_f^{-1}(\boldsymbol{g}^\sharp)$. Note that $\Psi_f^{-1}$ is a non-decreasing function with respect to $g_f$, so it holds that

$$g_f \ge g_f^\sharp.$$

Therefore, following the similar arguments as before, we can conclude that (26) holds, which gives the following result:

$$\Delta W_{\mathrm{OVA}}(\boldsymbol{g}; \boldsymbol{\eta}) \ge \inf_{g_f' = \theta} \Delta W_{\mathrm{OVA}}(\boldsymbol{g}; \boldsymbol{\eta})$$

$$\ge C^{-s}|\eta_f - (1 - c)|^s \qquad \text{(by assumption (15))}$$

$$\ge (2C)^{-s}|\eta_{y^*} - \eta_f|^s \qquad \text{(by assumption } c < \frac{1}{2})$$

$$\ge (2C)^{-s}\Delta W_{\text{0-1-}c}(r_f, f; \boldsymbol{\eta})^s. \qquad \text{(by using (27))}$$

Therefore the proof is completed. $\qquad\qquad\qquad\qquad\qquad\qquad\qquad\qquad\qquad\qquad \square$

### A.2 Derivation of $\theta$, $C$ and $s$ in Table 1

We now derive $\theta$, $C$ and $s$ in Table 1. The derivation goes along a similar discussion as Zhang [31], Yuan and Wegkamp [29], where they derived $C$ and $s$ in binary classification with rejection.

Define $\boldsymbol{g}^\dagger = \operatorname{argmin}_{g_y = \theta} W_{\mathrm{OVA}}(\boldsymbol{g}; \boldsymbol{\eta})$, and $\boldsymbol{g}^* = \operatorname{argmin}_{\boldsymbol{g}} W_{\mathrm{OVA}}(\boldsymbol{g}; \boldsymbol{\eta})$. Similarly to (22), we have

$$\frac{\phi'(-g_{y'}^\dagger)}{\phi'(-g_{y'}^\dagger) + \phi'(g_{y'}^\dagger)} = \begin{cases} \eta_{y'} & (y' \ne y), \\ 1 - c & (y' = y), \end{cases} \qquad (28)$$

$$\frac{\phi'(-g_{y'}^*)}{\phi'(-g_{y'}^*) + \phi'(g_{y'}^*)} = \eta_{y'} \qquad (y' \in \mathcal{Y}). \qquad (29)$$

By recalling the expression of $\Delta W_{\mathrm{OVA}}$, we see

$$\inf_{g_y = \theta} \Delta W_{\mathrm{OVA}}(\boldsymbol{g}; \boldsymbol{\eta}) = W_{\mathrm{OVA}}(\boldsymbol{g}^\dagger; \boldsymbol{\eta}) - W_{\mathrm{OVA}}(\boldsymbol{g}^*; \boldsymbol{\eta})$$

$$= \eta_y \phi(g_y^\dagger) + (1 - \eta_y)\phi(-g_y^\dagger) - \eta_y \phi(g_y^*) - (1 - \eta_y)\phi(-g_y^*). \qquad (30)$$

### A.2.1 Logistic loss

Observe that when we use logistic loss $\phi(z) = \log(1 + \exp(-z))$,

$$\frac{\phi'(-\theta)}{\phi'(-\theta) + \phi'(\theta)} = \frac{-\frac{1}{1+\exp(-\theta)}}{-\frac{1}{1+\exp(-\theta)} - \frac{1}{1+\exp(\theta)}} = \frac{1}{1 + \exp(-\theta)}.$$

Thus, we have (14) by letting $\theta = \log \frac{1-c}{c}$, which means that the requirement for $\phi$ in Theorem 7 is satisfied. By (28) and (29), we have $g_y^\dagger = \theta = \log \frac{1-c}{c}$ and $g_y^* = \log \frac{\eta_y}{1-\eta_y}$. Define the function $Q(t) = -t \log t - (1 - t) \log(1 - t)$. Then it holds that

$$
\begin{aligned}
\inf_{g_y=\theta} \Delta W_{\text{OVA}}(\boldsymbol{g}; \boldsymbol{\eta}) &= -\eta_y \log(1 - c) - (1 - \eta_y) \log c + \eta_y \log \eta_y + (1 - \eta_y) \log(1 - \eta_y) \\
&= -(1 - c) \log(1 - c) - c \log c + \eta_y \log \eta_y + (1 - \eta_y) \log(1 - \eta_y) \\
&\qquad + \big( \log c - \log(1 - c) \big) \big( \eta_y - (1 - c) \big) \\
&= Q(1 - c) - Q(\eta_y) + Q'(1 - c) \big( \eta_y - (1 - c) \big).
\end{aligned}
$$

By applying Taylor expansion to $Q$ at $t = 1 - c$, we know that there exists $\eta'$ between $\eta_y$ and $1 - c$ so that

$$
\begin{aligned}
\inf_{g_y=\theta} \Delta W_{\text{OVA}}(\boldsymbol{g}; \boldsymbol{\eta}) &= Q(1 - c) - Q(\eta_y) + Q'(1 - c) \big( \eta_y - (1 - c) \big) \\
&= -\frac{1}{2} Q''(\eta') \big( \eta_y - (1 - c) \big)^2 \\
&= \frac{1}{2\eta'(1 - \eta')} \big( \eta_y - (1 - c) \big)^2 \\
&\geq 2 \big( \eta_y - (1 - c) \big)^2,
\end{aligned}
$$

where the inequality follows from $\eta'(1 - \eta') \leq \frac{1}{4}$.

### A.2.2 Exponential loss

Observe that when we use exponential loss $\phi(z) = \exp(-z)$,

$$\frac{\phi'(-\theta)}{\phi'(-\theta) + \phi'(\theta)} = \frac{-\exp(\theta)}{-\exp(\theta) - \exp(-\theta)} = \frac{1}{1 + \exp(-2\theta)}.$$

Thus, we have (14) by letting $\theta = \frac{1}{2} \log \frac{1-c}{c}$, which means that the requirement for $\phi$ in Theorem 7 is satisfied. By (28) and (29), we have $g_y^\dagger = \theta = \frac{1}{2} \log \frac{1-c}{c}$ and $g_y^* = \frac{1}{2} \log \frac{\eta_y}{1-\eta_y}$. Define the function $Q(t) = 2\sqrt{t(1 - t)}$, then it holds that

$$
\begin{aligned}
\inf_{g_y=\theta} \Delta W_{\text{OVA}}(\boldsymbol{g}; \boldsymbol{\eta}) &= \eta_y \sqrt{\frac{c}{1 - c}} + (1 - \eta_y) \sqrt{\frac{1 - c}{c}} - 2\sqrt{\eta_y(1 - \eta_y)} \\
&= 2\sqrt{(1 - c)c} - 2\sqrt{\eta_y(1 - \eta_y)} + \frac{1 - 2(1 - c)}{\sqrt{(1 - c)c}} \big( \eta_y - (1 - c) \big) \\
&= Q(1 - c) - Q(\eta_y) + Q'(1 - c) \big( \eta_y - (1 - c) \big).
\end{aligned}
$$

Applying Taylor expansion to $Q$ at $t = 1 - c$, we know that there exists $\eta'$ between $\eta_y$ and $1 - c$ so that

$$
\begin{aligned}
\inf_{g_y=\theta} \Delta W_{\text{OVA}}(\boldsymbol{g}; \boldsymbol{\eta}) &= Q(1 - c) - Q(\eta_y) + Q'(1 - c) \big( \eta_y - (1 - c) \big) \\
&= -\frac{1}{2} Q''(\eta') \big( \eta_y - (1 - c) \big)^2 \\
&= \frac{1}{4} [\eta'(1 - \eta')]^{-\frac{3}{2}} \big( \eta_y - (1 - c) \big)^2 \\
&\geq 2 \big( \eta_y - (1 - c) \big)^2,
\end{aligned}
$$

where the inequality follows from $\eta'(1 - \eta') \leq \frac{1}{4}$.

### A.2.3 Squared loss

Observe that when we use squared loss $\phi(z) = (1-z)^2$, it holds that

$$\frac{\phi'(-\theta)}{\phi'(-\theta) + \phi'(\theta)} = \frac{2(-\theta-1)}{2(-\theta-1) + 2(\theta-1)} = \frac{1}{2}(\theta+1).$$

Thus, we have (14) by letting $\theta = 1 - 2c$, which means that the requirement for $\phi$ in Theorem 7 is satisfied. By (28) and (29), we have $g_y^\dagger = \theta = 1 - 2c$ and $g_y^* = 2\eta_y - 1$. Substitute these values into (30) gives

$$\inf_{g_y = \theta} \Delta W_{\mathrm{OVA}}(\boldsymbol{g}; \boldsymbol{\eta}) = 4\big(\eta_y - (1-c)\big)^2.$$

### A.2.4 Squared hinge loss

This is similar to the derivation of squared loss. Observe that when we use squared loss $\phi(z) = (1-z)_+^2$, it holds that

$$\frac{\phi'(-\theta)}{\phi'(-\theta) + \phi'(\theta)} = \frac{-2(1+\theta)_+}{-2(1+\theta)_+ - 2(1-\theta)_+} = \min\left\{1, \left(\frac{1}{2}(\theta+1)\right)_+\right\}.$$

Thus, we have (14) by letting $\theta = 1 - 2c$, which means that the requirement for $\phi$ in Theorem 7 is satisfied. By (28) and (29), we have $g_y^\dagger = \theta = 1 - 2c$ and $g_y^* = 2\eta_y - 1$. Together with the assumption $c < \frac{1}{2}$, it holds that

$$\begin{aligned}
\inf_{g_y = \theta} &\Delta W_{\mathrm{OVA}}(\boldsymbol{g}; \boldsymbol{\eta}) \\
&= \eta_y(1 - g_y^\dagger)_+^2 + (1 - \eta_y)(1 + g_y^\dagger)_+^2 - \eta_y(1 - g_y^*)_+^2 - (1 - \eta_y)(1 + g_y^*)_+^2 \\
&= 4c^2\eta_y + 4(1-c)^2(1 - \eta_y) - 4\eta_y(1 - \eta_y)^2 - 4\eta_y^2(1 - \eta_y) \\
&= 4(1-c)^2 + 4\eta_y(2c - 1) - 4\eta_y(1 - \eta_y) \\
&= 4\big(\eta_y - (1-c)\big)^2.
\end{aligned}$$

### A.3 Proof of the estimation error bound of OVA loss

For the case where only finite samples are available, we can bound the estimation error with respect to $0$-$1$-$c$ risk in the following way.

Let $\mathcal{G}$ be a family of functions $g_y : \mathcal{X} \to \mathbb{R}$. We denote by $\widehat{f}$ the minimizer over $\mathcal{G}$ of the empirical OVA risk $\widehat{R}(f) = \frac{1}{n} \sum_{i=1}^n \mathcal{L}_{\mathrm{OVA}}(f; \boldsymbol{x}_i, y_i)$, and its corresponding rejector (10) by $r_{\widehat{f}}$.

**Proposition 9** (Estimation error bound for OVA loss). *Assume $\phi$ is Lipschitz-continuous with constant $L_\phi$, and assume all functions in the model class $\mathcal{G}$ are bounded. Define $M_\phi = \sup_{\boldsymbol{x} \in \mathcal{X}, g \in \mathcal{G}} \phi\big(g(\boldsymbol{x})\big)$ and let $\mathfrak{R}_n(\mathcal{G})$ be the Rademacher complexity of $\mathcal{G}$ for data of size $n$ drawn from $p(\boldsymbol{x})$. Then under the assumption of Theorem 7, for any $\delta > 0$, it holds with probability at least $1 - \delta$ that*

$$\Delta R_{\text{0-1-}c}(r_{\widehat{f}}, \widehat{f}) \leq 2C \left( \inf_{g_1, \dots, g_K \in \mathcal{G}} \Delta R_{\mathrm{OVA}}(f) + 8KL_\phi \mathfrak{R}_n(\mathcal{G}) + 4M_\phi \sqrt{\frac{2 \log \frac{2}{\delta}}{n}} \right)^{\frac{1}{s}}.$$

This proposition is straightforward from Theorem 7 and the standard argument regarding Rademacher complexity [18].

**Definition 10** (Rademacher Complexity [18]). Let $Z_1, \dots, Z_n$ be random variables drawn i.i.d. from a probability distribution $D$, and $\mathcal{H} = \{h : \mathcal{Z} \to \mathbb{R}\}$ be a family of measurable function. Then the Rademacher complexity of $\mathcal{H}$ is defined as

$$\mathfrak{R}_n(\mathcal{H}) = \mathop{\mathbb{E}}_{Z_1, \dots, Z_n \sim D} \mathop{\mathbb{E}}_{\sigma_1, \dots \sigma_n} \left[ \sup_{h \in \mathcal{H}} \frac{1}{n} \sum_{i=1}^n \sigma_i h(Z_i) \right],$$

where $\sigma_1, \dots, \sigma_n$ are Rademacher variables, which are independent uniform random variables taking values in $\{-1, +1\}$.

We first show the following lemmas. Define $\mathcal{H}_{\mathrm{OVA}}$ as

$$\mathcal{H}_{\mathrm{OVA}} = \{(\boldsymbol{x}, y) \mapsto \mathcal{L}_{\mathrm{OVA}}(\boldsymbol{g}; \boldsymbol{x}, y) \mid g_1, \ldots, g_K \in \mathcal{G}\}.$$

**Lemma 11.** *Let $\mathfrak{R}_n(\mathcal{H}_{\mathrm{OVA}})$ be the Rademacher complexity of $\mathcal{H}_{\mathrm{OVA}}$ for $\mathcal{S} = \{(\boldsymbol{x}_i, y_i)\}_{i=1}^{n}$ from $p(\boldsymbol{x}, y)$, and $\mathfrak{R}_n(\mathcal{G})$ be the Rademacher complexity of $\mathcal{G}$ for data of size $n$ drawn from $p(\boldsymbol{x})$. Then we have $\mathfrak{R}_n(\mathcal{H}_{\mathrm{OVA}}) \leq 2KL_\phi \mathfrak{R}_n(\mathcal{G})$.*

*Proof of Lemma 11.* By definition, we can bound $\mathfrak{R}_n(\mathcal{H}_{\mathrm{OVA}})$ as follows:

$$\mathfrak{R}_n(\mathcal{H}_{\mathrm{OVA}}) = \mathbb{E}_{\mathcal{S}} \mathbb{E}_{\sigma_1,\ldots,\sigma_n} \left[ \sup_{g_1,\ldots,g_K \in \mathcal{G}} \frac{1}{n} \sum_{(\boldsymbol{x}_i,y_i) \in \mathcal{S}} \sigma_i \left( \phi\big(g_{y_i}(\boldsymbol{x}_i)\big) + \sum_{y' \neq y_i} \phi\big(-g_{y'}(\boldsymbol{x}_i)\big) \right) \right]$$

$$\leq \underbrace{\mathbb{E}_{\mathcal{S}} \mathbb{E}_{\sigma_1,\ldots,\sigma_n} \left[ \sup_{g_1,\ldots,g_K \in \mathcal{G}} \frac{1}{n} \sum_{(\boldsymbol{x}_i,y_i) \in \mathcal{S}} \sigma_i \phi\big(g_{y_i}(\boldsymbol{x}_i)\big) \right]}_{(\mathrm{A})}$$

$$+ \underbrace{\mathbb{E}_{\mathcal{S}} \mathbb{E}_{\sigma_1,\ldots,\sigma_n} \left[ \sup_{g_1,\ldots,g_K \in \mathcal{G}} \frac{1}{n} \sum_{(\boldsymbol{x}_i,y_i) \in \mathcal{S}} \sigma_i \sum_{y' \neq y_i} \phi\big(-g_{y'}(\boldsymbol{x}_i)\big) \right]}_{(\mathrm{B})}.$$

where we utilized the sub-additivity of supremum. We shall bound the both terms above. By letting $\alpha_i = 2\mathbb{1}_{[y=y_i]} - 1$, we can bound the first term (A) as follows:

$$(\mathrm{A}) = \mathbb{E}_{\mathcal{S}} \mathbb{E}_{\sigma_1,\ldots,\sigma_n} \left[ \sup_{g_1,\ldots,g_K \in \mathcal{G}} \frac{1}{n} \sum_{(\boldsymbol{x}_i,y_i) \in \mathcal{S}} \sigma_i \phi\big(g_{y_i}(\boldsymbol{x}_i)\big) \right]$$

$$= \mathbb{E}_{\mathcal{S}} \mathbb{E}_{\sigma_1,\ldots,\sigma_n} \left[ \sup_{g_1,\ldots,g_K \in \mathcal{G}} \frac{1}{2n} \sum_{(\boldsymbol{x}_i,y_i) \in \mathcal{S}} \sigma_i \sum_{y} \phi\big(g_y(\boldsymbol{x}_i)\big)(\alpha_i + 1) \right]$$

$$\leq \mathbb{E}_{\mathcal{S}} \mathbb{E}_{\sigma_1,\ldots,\sigma_n} \left[ \sup_{g_1,\ldots,g_K \in \mathcal{G}} \frac{1}{2n} \sum_{(\boldsymbol{x}_i,y_i) \in \mathcal{S}} \alpha_i \sigma_i \sum_{y} \phi\big(g_y(\boldsymbol{x}_i)\big) \right]$$

$$+ \mathbb{E}_{\mathcal{S}} \mathbb{E}_{\sigma_1,\ldots,\sigma_n} \left[ \sup_{g_1,\ldots,g_K \in \mathcal{G}} \frac{1}{2n} \sum_{(\boldsymbol{x}_i,y_i) \in \mathcal{S}} \sigma_i \sum_{y} \phi\big(g_y(\boldsymbol{x}_i)\big) \right]$$

$$= \mathbb{E}_{\mathcal{S}} \mathbb{E}_{\sigma_1,\ldots,\sigma_n} \left[ \sup_{g_1,\ldots,g_K \in \mathcal{G}} \frac{1}{n} \sum_{(\boldsymbol{x}_i,y_i) \in \mathcal{S}} \sigma_i \sum_{y} \phi\big(g_y(\boldsymbol{x}_i)\big) \right] \tag{31}$$

$$\leq \sum_{y} \mathbb{E}_{\mathcal{S}} \mathbb{E}_{\sigma_1,\ldots,\sigma_n} \left[ \sup_{g_1,\ldots,g_K \in \mathcal{G}} \frac{1}{n} \sum_{(\boldsymbol{x}_i,y_i) \in \mathcal{S}} \sigma_i \phi\big(g_y(\boldsymbol{x}_i)\big) \right]$$

$$= K\mathfrak{R}_n(\phi \circ \mathcal{G}).$$

We utilized the sub-additivity again and in (31), we used the fact that $\alpha_i \sigma_i$ has exactly the same distribution as $\sigma_i$.

Similarly to term (A), the second term (B) can be bounded by $K\mathfrak{R}_n(\phi \circ \mathcal{G})$.

Consequently, we can bound the Rademacher complexity of $\mathcal{H}_{\mathrm{OVA}}$ as follows:

$$\mathfrak{R}_n(\mathcal{H}_{\mathrm{OVA}}) \leq 2K\mathfrak{R}_n(\phi \circ \mathcal{G}) \leq 2KL_\phi \mathfrak{R}_n(\mathcal{G})$$

due to Talagrand's contraction lemma. $\qquad\square$

**Lemma 12.** *For any $\delta$, with probability at least $1 - \delta$,*

$$\sup_{g_1,\ldots,g_K \in \mathcal{G}} \left| \widehat{R}_{\mathrm{OVA}}(\boldsymbol{g}) - R_{\mathrm{OVA}}(\boldsymbol{g}) \right| \leq 4KL_\phi \mathfrak{R}(\mathcal{G}) + M_\phi \sqrt{\frac{8 \log \frac{2}{\delta}}{n}}.$$

*Proof of Lemma 12.* We will only discuss a one-sided bound on $\sup_{g_1,\dots,g_K \in \mathcal{G}} \big(\widehat{R}_{\text{OVA}}(\boldsymbol{g}) - R_{\text{OVA}}(\boldsymbol{g})\big)$ that holds with probability at least $1 - \frac{\delta}{2}$. The other side can be derived in a similar way.

To begin with, we first bound the change of $\sup_{g_1,\dots,g_K \in \mathcal{G}} \big(\widehat{R}_{\text{OVA}}(\boldsymbol{g}) - R_{\text{OVA}}(\boldsymbol{g})\big)$ when a single entry $z_i = (\boldsymbol{x}_i, y_i)$ of $(z_i, \dots, z_n)$ is replaced with $z_i' = (\boldsymbol{x}_i', y_i')$. Define $A(z_1, \dots, z_n) = \sup_{g_1,\dots,g_K \in \mathcal{G}} \big(\widehat{R}_{\text{OVA}}(\boldsymbol{g}) - R_{\text{OVA}}(\boldsymbol{g})\big)$. Then it holds that

$$A(z_1, \dots, z_i, \dots, z_n) - A(z_1, \dots, z_i', \dots, z_n)$$

$$= \sup_{g_1,\dots,g_K \in \mathcal{G}} \inf_{g_1',\dots,g_K' \in \mathcal{G}} \left[ \frac{1}{n} \sum_{j=1}^{n} \mathcal{L}_{\text{OVA}}(\boldsymbol{g}; \boldsymbol{x}_j, y_j) - \mathbb{E}_{p(\boldsymbol{x},y)} [\mathcal{L}_{\text{OVA}}(\boldsymbol{g}; \boldsymbol{x}, y)] \right.$$

$$\left. - \frac{1}{n} \sum_{j \in \{1,\dots,n\} \setminus \{i\}} \mathcal{L}_{\text{OVA}}(\boldsymbol{g}'; \boldsymbol{x}_j, y_j) - \frac{1}{n} \mathcal{L}_{\text{OVA}}(\boldsymbol{g}'; \boldsymbol{x}_i', y_i') + \mathbb{E}_{p(\boldsymbol{x},y)} [\mathcal{L}_{\text{OVA}}(\boldsymbol{g}'; \boldsymbol{x}, y)] \right]$$

$$\leq \sup_{g_1,\dots,g_K \in \mathcal{G}} \left[ \frac{1}{n} \sum_{j=1}^{n} \mathcal{L}_{\text{OVA}}(\boldsymbol{g}; \boldsymbol{x}_j, y_j) - \mathbb{E}_{p(\boldsymbol{x},y)} [\mathcal{L}_{\text{OVA}}(\boldsymbol{g}; \boldsymbol{x}, y)] \right.$$

$$\left. - \frac{1}{n} \sum_{j \in \{1,\dots,n\} \setminus \{i\}} \mathcal{L}_{\text{OVA}}(\boldsymbol{g}; \boldsymbol{x}_j, y_j) - \frac{1}{n} \mathcal{L}_{\text{OVA}}(\boldsymbol{g}; \boldsymbol{x}_i', y_i') + \mathbb{E}_{p(\boldsymbol{x},y)} [\mathcal{L}_{\text{OVA}}(\boldsymbol{g}; \boldsymbol{x}, y)] \right]$$

$$= \sup_{g_1,\dots,g_K \in \mathcal{G}} \left[ \frac{1}{n} \mathcal{L}_{\text{OVA}}(\boldsymbol{g}; \boldsymbol{x}_i, y_i) - \frac{1}{n} \mathcal{L}_{\text{OVA}}(\boldsymbol{g}; \boldsymbol{x}_i', y_i') \right]$$

$$= \frac{1}{n} \sup_{g_1,\dots,g_K \in \mathcal{G}} \left[ \phi\big(g_{y_i}(\boldsymbol{x}_i)\big) + \sum_{y' \neq y_i} \phi\big(-g_{y'}(\boldsymbol{x}_i)\big) - \phi\big(g_{y_i'}(\boldsymbol{x}_i')\big) - \sum_{y'' \neq y_i'} \phi\big(-g_{y''}(\boldsymbol{x}_i')\big) \right]$$

$$= \frac{1}{n} \sup_{g_1,\dots,g_K \in \mathcal{G}} \left[ \phi\big(g_{y_i}(\boldsymbol{x}_i)\big) + \phi\big(-g_{y_i'}(\boldsymbol{x}_i)\big) - \phi\big(g_{y_i'}(\boldsymbol{x}_i')\big) - \phi\big(-g_{y_i}(\boldsymbol{x}_i')\big) \right] \leq \frac{4M_\phi}{n},$$

where we used the boundedness of the loss function.

Thus, we can apply McDiarmid's inequality to get that with probability at least $1 - \frac{\delta}{2}$,

$$\sup_{g_1,\dots,g_K \in \mathcal{G}} \big(\widehat{R}_{\text{OVA}}(\boldsymbol{g}) - R_{\text{OVA}}(\boldsymbol{g})\big) \leq \mathbb{E}\left[ \sup_{g_1,\dots,g_K \in \mathcal{G}} \big(\widehat{R}_{\text{OVA}}(\boldsymbol{g}) - R_{\text{OVA}}(\boldsymbol{g})\big) \right] + M_\phi \sqrt{\frac{8 \log \frac{2}{\delta}}{n}}.$$

Since $R_{\text{OVA}}(\boldsymbol{g}) = \mathbb{E}\big[\widehat{R}_{\text{OVA}}(\boldsymbol{g})\big]$, thus by applying symmetrization [18], we get

$$\mathbb{E}\left[ \sup_{g_1,\dots,g_K \in \mathcal{G}} \big(\widehat{R}_{\text{OVA}}(\boldsymbol{g}) - R_{\text{OVA}}(\boldsymbol{g})\big) \right] \leq 2\mathfrak{R}_n(\mathcal{H}_{\text{OVA}}) \leq 2KL_\phi \mathfrak{R}_n(\mathcal{G}),$$

where the last line inequality follows from Lemma 11. $\qquad\square$

Lastly, we present the proof for Proposition 9.

*Proof of Proposition 9.* To begin with, we split the excess risk into two parts: the estimation error term and the approximation term as follow:

$$R(\widehat{\boldsymbol{g}}) - R^* = \underbrace{\left( R(\widehat{\boldsymbol{g}}) - \inf_{g_1,\dots,g_K \in \mathcal{G}} R(\boldsymbol{g}) \right)}_{\text{estimation error}} + \underbrace{\left( \inf_{g_1,\dots,g_K \in \mathcal{G}} R(\boldsymbol{g}) - R^* \right)}_{\text{approximation error}}.$$

We focus on the estimation error. Therefore, we assume that for any $\varepsilon > 0$, there exist $g_1^\circ, \dots, g_K^\circ \in \mathcal{G}$ such that

$$R(\boldsymbol{g}^\circ) = \inf_{g_1,\dots,g_K \in \mathcal{G}} R(\boldsymbol{g}) - \varepsilon.$$

Figure 3: The comparison of the calibration function using the Cross Entropy loss. The blue line corresponds to the case of ordinary classification, and the red line corresponds to the case of learning with rejection, which is our framework. As we can see from the graph, both functions show similar behavior near $z \simeq 0$.

Using these notations, we can upper bound the estimation error as follows:

$$
\begin{aligned}
R(\widehat{\boldsymbol{g}}) - &\inf_{g_1,\ldots,g_K \in \mathcal{G}} R(\boldsymbol{g}) \\
&= R(\widehat{\boldsymbol{g}}) - R(\boldsymbol{g}^\circ) \\
&= \left( R(\widehat{\boldsymbol{g}}) - \widehat{R}(\widehat{\boldsymbol{g}}) \right) + \left( \widehat{R}(\boldsymbol{g}^\circ) - R(\boldsymbol{g}^\circ) \right) + \left( \widehat{R}(\widehat{\boldsymbol{g}}) - \widehat{R}(\boldsymbol{g}^\circ) \right) \\
&\leq 2 \sup_{g_1,\ldots,g_K \in \mathcal{G}} \left| \widehat{R}(\boldsymbol{g}) - R(\boldsymbol{g}) \right| + \left( \widehat{R}(\widehat{\boldsymbol{g}}) - \widehat{R}(\boldsymbol{g}^\circ) \right) \\
&\leq 2 \sup_{g_1,\ldots,g_K \in \mathcal{G}} \left| \widehat{R}(\boldsymbol{g}) - R(\boldsymbol{g}) \right| \qquad\qquad (32) \\
&\leq 4 K L_\phi \mathfrak{R}(\mathcal{G}) + M_\phi \sqrt{\frac{8 \log \frac{2}{\delta}}{n}}, \qquad\qquad (33)
\end{aligned}
$$

where we used that $\widehat{R}(\widehat{\boldsymbol{g}}) \leq \widehat{R}(\boldsymbol{g}^\circ)$ by the definition of $\widehat{\boldsymbol{g}}$ in (32), and we used the result of Lemma 11 in (33). The above result together with Theorem 7 gives Proposition 9. $\qquad\square$

### A.4 Proof of Theorem 8

To begin with, we will use the following theorem, which is proved in Pires and Szepesvári [19].

**Theorem 13** (Pires and Szepesvári [19]). *Define the function $\xi_{\mathrm{CE}} : \mathbb{R} \to \mathbb{R}_{\geq 0}$ as*

$$
\xi_{\mathrm{CE}}(z) = \frac{1}{2} \big[ (1+z)\log(1+z) + (1-z)\log(1-z) \big].
$$

*Then for all $f$, we have*

$$
\xi_{\mathrm{CE}}\big( \Delta R_{0\text{-}1}(f) \big) \leq \Delta R_{\mathrm{CE}}(\boldsymbol{g}).
$$

To compare Theorem 8 with Theorem 13, let us apply Taylor expansion to calibration function $\xi_{\mathrm{CE}}$:

$$
\xi_{\mathrm{CE}}(z) = \frac{1}{2} z^2 + \frac{1}{12} z^4 + \frac{1}{30} z^6 + \mathrm{O}(z^8) > \frac{1}{2} z^2.
$$

As we can see from the above, Theorem 8 provides a loosened excess risk bound compared to Theorem 13. Yet, we can observe that the behavior of $\xi_{\mathrm{CE}}$ is similar to that of $\frac{1}{2} z^2$ when $z \in [0, 1]$ (see also Figure 3), thus gives similar calibration performance.

Now we will prove Theorem 8.

*Proof.* The pointwise risk of CE loss is expressed as

$$W_{\mathrm{CE}}(\boldsymbol{g};\boldsymbol{\eta}) = -\sum_y \eta_y g_y + \log\left(\sum_y \exp(g_y)\right). \tag{34}$$

Similarly to the proof of Theorem 7, the main focus in this proof is to show the following inequality:

$$\frac{1}{2}\Delta W_{\text{0-1-}c}(r_f, f; \boldsymbol{\eta})^2 \le \Delta W_{\mathrm{CE}}(\boldsymbol{g}; \boldsymbol{\eta}).$$

Note that when we use cross entropy loss, we can rewrite the surrogate excess risk using KL divergence $\mathcal{D}_{\mathrm{KL}}(\cdot \parallel \cdot)$

$$
\begin{aligned}
\Delta W_{\mathrm{CE}}(\boldsymbol{g};\boldsymbol{\eta}) &= -\sum_y \eta_y g_y + \log\left(\sum_y \exp(g_y)\right) + \sum_y \eta_y \log \eta_y \qquad \text{(by definition (34))}\\
&= -\sum_y \eta_y \log \Psi_y^{-1}(\boldsymbol{g}) + \sum_y \eta_y \log \eta_y \\
&= \sum_y \eta_y \log \frac{\eta_y}{\Psi_y^{-1}(\boldsymbol{g})} = \mathcal{D}_{\mathrm{KL}}(\boldsymbol{\eta} \parallel \boldsymbol{\Psi}^{-1}(\boldsymbol{g}))
\end{aligned}
$$

Now we us Table 5 again. Note that $\Psi_f(\boldsymbol{x}) = \max_{y \in \mathcal{Y}} \Psi_y(\boldsymbol{x})$ if we use proper composite loss.

**Cases (A)(G)(H):**

In this case, we can confirm that $\Delta W_{\text{0-1-}c}(r_f, f; \boldsymbol{\eta}) = 0$ by (17) since $(r_f, f)$ makes a correct prediction. Thus, it holds that

$$\Delta W_{\text{0-1-}c}(r_f, f; \boldsymbol{\eta}) = 0 \le \sqrt{2\Delta W_{\mathrm{CE}}(\boldsymbol{g}; \boldsymbol{\eta})}.$$

**Cases (C)(D):**

In this case, we can confirm that $\Delta W_{\text{0-1-}c}(r_f, f; \boldsymbol{\eta}) = c - (1 - \eta_{y^*})$ by (17), thus, we can lower bound surrogate excess risk as follows:

$$
\begin{aligned}
\Delta W_{\mathrm{CE}}(\boldsymbol{g};\boldsymbol{\eta}) &= \mathcal{D}_{\mathrm{KL}}(\boldsymbol{\eta} \parallel \boldsymbol{\Psi}^{-1}(\boldsymbol{g})) \\
&\ge \frac{1}{2}\left(\sum_y \left|\eta_y - \Psi_y^{-1}(\boldsymbol{g})\right|\right)^2 \qquad \text{(Pinsker's inequality)} \\
&\ge \frac{1}{2}\left|\eta_{y^*} - \Psi_{y^*}^{-1}(\boldsymbol{g})\right|^2 \\
&\ge \frac{1}{2}\left|\eta_{y^*} - (1-c)\right|^2 \tag{35} \\
&= \frac{1}{2}\Delta W_{\text{0-1-}c}(r_f, f; \boldsymbol{\eta})^2,
\end{aligned}
$$

where in (35), we used the fact that $\eta_{y^*} > 1 - c$ in cases (C)(D), and $\Psi_{y^*}^{-1}(\boldsymbol{g}) \le \Psi_{y^*}^{-1}(\boldsymbol{g})$ by $\max_y \Psi_y^{-1}(\boldsymbol{g}) = \Psi_f^{-1}(\boldsymbol{g}) \le 1 - c$.

**Cases (E)(F):**

In this case, we can confirm that $\Delta W_{\text{0-1-}c}(r_f, f; \boldsymbol{\eta}) = |(1 - \eta_f) - c|$ by (17), thus, similar to the case above, we can lower bound surrogate excess risk as follows:

$$
\begin{aligned}
\Delta W_{\mathrm{CE}}(\boldsymbol{g};\boldsymbol{\eta}) &= \mathcal{D}_{\mathrm{KL}}(\boldsymbol{\eta} \parallel \boldsymbol{\Psi}^{-1}(\boldsymbol{g})) \\
&\ge \frac{1}{2}\left(\sum_y \left|\eta_y - \Psi_y^{-1}(\boldsymbol{g})\right|\right)^2 \qquad \text{(Pinsker's inequality)} \\
&\ge \frac{1}{2}\left|\eta_f - \Psi_f^{-1}(\boldsymbol{g})\right|^2 \\
&\ge \frac{1}{2}\left|\eta_f - (1-c)\right|^2 \qquad \text{(by } \eta_f \le 1 - c < \Psi_f^{-1}(\boldsymbol{g}) \text{ in cases (E)(F))} \\
&= \frac{1}{2}\Delta W_{\text{0-1-}c}(r_f, f; \boldsymbol{\eta})^2.
\end{aligned}
$$

**Case (B):**

This case reduces to the excess risk bound in the ordinary classification, which enables us to utilize the result of Theorem 13. Note that $\Delta W_{0\text{-}1\text{-}c}(r_f, f; \boldsymbol{\eta}) = \Delta W_{0\text{-}1}(f; \boldsymbol{\eta}) = \eta_{y^*} - \eta_f$ by (17). Thus, we can lower bound $\Delta W_{\mathrm{CE}}(\boldsymbol{g}; \boldsymbol{\eta})$ as follows:

$$\Delta W_{\mathrm{CE}}(\boldsymbol{g}; \boldsymbol{\eta}) \geq \xi_{\mathrm{CE}}\big(\Delta W_{0\text{-}1}(f; \boldsymbol{\eta})\big) = \xi_{\mathrm{CE}}\big(\Delta W_{0\text{-}1\text{-}c}(r_f, f; \boldsymbol{\eta})\big) \geq \frac{1}{2}\Delta W_{0\text{-}1\text{-}c}(r_f, f; \boldsymbol{\eta})^2,$$

where in the last inequality, we used the property $\xi_{\mathrm{CE}}(z) \geq \frac{1}{2}z^2$.

Combining the above analysis completes the proof. $\qquad\square$

# B Analysis for classifier-rejector approach

## B.1 Proof of Theorem 4

Define $h_{\boldsymbol{\eta}}(r) = \frac{\partial W(r, f_{\boldsymbol{\eta}}^\dagger; \boldsymbol{\eta})}{\partial r}$. Note that the minimizer $r_{\boldsymbol{\eta}}^\dagger$ in (3) satisfies the first-order optimality condition

$$h_{\boldsymbol{\eta}}(r_{\boldsymbol{\eta}}^\dagger) = 0. \tag{36}$$

We first consider the case $\max_y \eta_y \leq 1 - c$, i.e., $\mathrm{sign}[r^*] = -1$. Recall that $W$ is defined as

$$W\big(r(\boldsymbol{x}), f(\boldsymbol{x}); \boldsymbol{\eta}(\boldsymbol{x})\big) = \sum_{y \in \mathcal{Y}} \eta_y(\boldsymbol{x})\mathcal{L}\big(r, f; \boldsymbol{x}, y\big)$$

is a convex combination of $\mathcal{L}$, which is a convex function of class $C^1$. Thus, $h_{\boldsymbol{\eta}}(r)$ is a non-decreasing function with respect to $r$. Since we assumed that $r^\dagger$ is rejection-calibrated, we need $\mathrm{sign}[r_{\boldsymbol{\eta}}^\dagger] = \mathrm{sign}[r^*] = -1$, which implies $r_{\boldsymbol{\eta}}^\dagger < 0$. Therefore we have $h_{\boldsymbol{\eta}}(0) \geq 0$ by the monotonicity and (36). Since this argument holds for any $\boldsymbol{\eta}$ such that $\max_y \eta_y \leq 1-c$, we have $\sup_{\boldsymbol{\eta}: \max_y \eta_y \leq 1-c} h_{\boldsymbol{\eta}}(0) \geq 0$. For the case $\max_y \eta_y \geq 1 - c$ we have $\inf_{\boldsymbol{\eta}: \max_y \eta_y \geq 1-c} h_{\boldsymbol{\eta}}(0) \leq 0$.

Combining the above analysis, we can conclude that $r^\dagger$ is rejection-calibrated only if

$$\sup_{\boldsymbol{\eta}: \max_y \eta_y \geq 1-c} h_{\boldsymbol{\eta}}(0) \leq 0 \leq \inf_{\boldsymbol{\eta}: \max_y \eta_y \leq 1-c} h_{\boldsymbol{\eta}}(0). \tag{37}$$

The necessary conditions (6) (left) and (6) (right) are then straightforward, since restricting constraints does not make the supremum larger and the infimum smaller. We further show in Appendix B.4 that (37) is also the sufficient condition for rejection calibration. $\qquad\square$

## B.2 Upper bounds of 0-1-$c$ loss

We first present general upper bounds of 0-1-$c$ loss in the multiclass setting.

**Lemma 14** (Upper bounds for 0-1-$c$ loss). *Let $\phi(z), \psi(z), \psi_1(z), \psi_2(z)$ be convex functions that bound $\mathbb{1}_{[z \leq 0]}$ from above, and $\alpha, \beta$ be the positive constants. Then, Additive Pairwise Comparison Loss (APC loss) $\mathcal{L}_{\mathrm{APC}}$ and Multiplicative Pairwise Comparison Loss (MPC loss) $\mathcal{L}_{\mathrm{MPC}}$ given below are upper bounds of 0-1-$c$ loss, where*

$$\mathcal{L}_{\mathrm{APC}}(r, f; \boldsymbol{x}, y) = \sum_{y' \neq y} \phi\Big(\alpha\big(g_y(\boldsymbol{x}) - g_{y'}(\boldsymbol{x}) - r(\boldsymbol{x})\big)\Big) + c\psi\big(\beta r(\boldsymbol{x})\big),$$

$$\mathcal{L}_{\mathrm{MPC}}(r, f; \boldsymbol{x}, y) = \sum_{y' \neq y} \phi\Big(\alpha\big(g_y(\boldsymbol{x}) - g_{y'}(\boldsymbol{x})\big)\Big)\psi_1(-\alpha r(\boldsymbol{x})) + c\psi_2\big(\beta r(\boldsymbol{x})\big).$$

*Proof.* Define the margin function: $m(f(\boldsymbol{x}), y) = g_y(\boldsymbol{x}) - \max_{y' \neq y} g_{y'}(\boldsymbol{x})$. Note that the margin function satisfies

$$\mathbb{1}_{[f(\boldsymbol{x}) \neq y]} \leq \mathbb{1}_{[m(f(\boldsymbol{x}), y) < 0]},$$
$$\phi\Big(\alpha m(f(\boldsymbol{x}), y)\Big) \leq \sum_{y' \neq y} \phi\Big(\alpha(g_y(\boldsymbol{x}) - g_{y'}(\boldsymbol{x}))\Big).$$

Using these properties, we can bound $\mathcal{L}_{0\text{-}1\text{-}c}$ from above as follows:

$$
\begin{aligned}
\mathcal{L}_{0\text{-}1\text{-}c}(r,f;\boldsymbol{x},y) &\leq \mathbb{1}_{[m(f(\boldsymbol{x}),y)<0]}\mathbb{1}_{[r(\boldsymbol{x})>0]} + c\mathbb{1}_{[r(\boldsymbol{x})\leq 0]} \\
&= \mathbb{1}_{[\alpha m(f(\boldsymbol{x}),y)<0]}\mathbb{1}_{[\alpha r(\boldsymbol{x})>0]} + c\mathbb{1}_{[\beta r(\boldsymbol{x})\leq 0]} \\
&\leq \phi\Big(\alpha m\left(f(\boldsymbol{x}),y\right)\Big)\psi_1(\alpha r(\boldsymbol{x})) + c\psi_2(\beta r(\boldsymbol{x})) \\
&\leq \sum_{y'\neq y}\phi\Big(\alpha(g_y(\boldsymbol{x})-g_{y'}(\boldsymbol{x}))\Big)\psi_1(\alpha r(\boldsymbol{x})) + c\psi_2(\beta r(\boldsymbol{x})) \\
&= \mathcal{L}_{\mathrm{MPC}}(r,f;\boldsymbol{x},y).
\end{aligned}
$$

$$
\begin{aligned}
\mathcal{L}_{0\text{-}1\text{-}c}(r,f;\boldsymbol{x},y) &\leq \mathbb{1}_{[m(f(\boldsymbol{x}),y)<0]}\mathbb{1}_{[r(\boldsymbol{x})>0]} + c\mathbb{1}_{[r(\boldsymbol{x})\leq 0]} \\
&\leq \mathbb{1}_{[m(f(\boldsymbol{x}),y)<0]}\mathbb{1}_{[-r(\boldsymbol{x})\leq 0]} + c\mathbb{1}_{[r(\boldsymbol{x})\leq 0]} \\
&= \mathbb{1}_{[\max(m(f(\boldsymbol{x}),y),-r(\boldsymbol{x}))\leq 0]} + c\mathbb{1}_{[r(\boldsymbol{x})\leq 0]} \\
&\leq \mathbb{1}_{[\frac{1}{2}(m(f(\boldsymbol{x}),y)-r(\boldsymbol{x}))\leq 0]} + c\mathbb{1}_{[r(\boldsymbol{x})\leq 0]} \\
&= \mathbb{1}_{[\alpha(m(f(\boldsymbol{x}),y)-r(\boldsymbol{x}))\leq 0]} + c\mathbb{1}_{[\beta r(\boldsymbol{x})\leq 0]} \\
&\leq \phi\Big(\alpha\left(m(f(\boldsymbol{x}),y)-r(\boldsymbol{x})\right)\Big) + c\psi(\beta r(\boldsymbol{x})) \\
&\leq \mathcal{L}_{\mathrm{APC}}(r,f;\boldsymbol{x},y).
\end{aligned}
$$

$\square$

In terms of optimization, $\mathcal{L}_{\mathrm{APC}}$ is convex with respect to $(r,f)$, while $\mathcal{L}_{\mathrm{MPC}}$ is non-convex since it contains the multiplication of two convex functions, which is not necessarily convex. However, we can easily confirm that $\mathcal{L}_{\mathrm{MPC}}$ is biconvex, that is, if we fix either $r$ or $f$, then this function is convex with respect to the other.

### B.3  Order-preserving property

We first show that these losses in Lemma 14 have the order-preserving property which is defined as follows.

**Definition 15** (Order-preserving property [30]). *A loss $\mathcal{L}$ is order-preserving if, for all fixed $\boldsymbol{\eta}$, its pointwise risk $W$ has a minimizer $\boldsymbol{g}^* \in \Omega$ such that $\eta_i < \eta_j \Rightarrow g_i^* \leq g_j^*$. Moreover, the loss $\mathcal{L}$ is strictly order-preserving if the minimizer $\boldsymbol{g}^*$ satisfies $\eta_i < \eta_j \Rightarrow g_i^* < g_j^*$.*

It is known that the order-preserving property is a sufficient condition for classification calibration [19]. Therefore, showing the order-preserving property of a loss function guarantees classification calibration.

Again, we denote by $(r^\dagger, f^\dagger)$ the minimizer of the above risks over all measurable functions, and $(r_{\boldsymbol{\eta}}^\dagger, f_{\boldsymbol{\eta}}^\dagger)$ the minimizer of the corresponding pointwise risks over real space:

$$
\begin{aligned}
(r^\dagger, f^\dagger) &= \operatorname*{argmin}_{r,f:\text{measurable}} R(r,f), \\
(r_{\boldsymbol{\eta}}^\dagger, f_{\boldsymbol{\eta}}^\dagger) &:= \operatorname*{argmin}_{r\in\mathbb{R},\, \boldsymbol{g}\in\mathbb{R}^K} W(r,f;\boldsymbol{\eta}),
\end{aligned}
$$

where we consider APC loss and MPC loss for the pointwise risk $W$, which are expressed as

$$
W_{\mathrm{APC}}(r,f;\boldsymbol{\eta}) = \sum_y \eta_y\left(\sum_{y'\neq y}\phi\Big(\alpha\big(g_y-g_{y'}-r\big)\Big)\right) + c\psi\big(\beta r\big),
$$

$$
W_{\mathrm{MPC}}(r,f;\boldsymbol{\eta}) = \sum_y \eta_y\left(\sum_{y'\neq y}\phi\Big(\alpha\big(g_y-g_{y'}\big)\Big)\psi_1(-\alpha r)\right) + c\psi_2\big(\beta r\big).
$$

The following theorems show that APC loss an MPC loss have order-preserving property.

**Theorem 16** (Order-preserving property for $\mathcal{L}_{\mathrm{APC}}$). *$\mathcal{L}_{\mathrm{APC}}$ is order-preserving if $\phi$ is a non-increasing function such that $\phi(z - \alpha r_{\boldsymbol{\eta}}^{\dagger}) < \phi(-z - \alpha r_{\boldsymbol{\eta}}^{\dagger})$ $(z > 0)$ holds for all $\boldsymbol{\eta} \in \Lambda_K$. Moreover, $\mathcal{L}_{\mathrm{APC}}$ is strictly order-preserving if $\phi$ is differentiable and $\phi'(-\alpha r_{\boldsymbol{\eta}}^{\dagger}) < 0$ holds for all $\boldsymbol{\eta} \in \Lambda_K$.*

**Theorem 17** (Order-preserving property for $\mathcal{L}_{\mathrm{MPC}}$). *$\mathcal{L}_{\mathrm{MPC}}$ is order-preserving if $\phi$ is a non-increasing function such that $\phi(z) < \phi(-z)$ holds for all $z > 0$. Moreover, $\mathcal{L}_{\mathrm{MPC}}$ is strictly order-preserving if $\phi$ is differentiable and $\phi'(0) < 0$.*

*Proof.* We will only prove Theorem 16. The proof of Theorem 17 proceeds along the same line as the proof of Theorem 16 and is thus omitted.

We can fix $i = 1, j = 2$ without loss of generality. Define $g'_{\boldsymbol{\eta},k}$ as

$$
g'_{\boldsymbol{\eta},k} = \begin{cases} g^{\dagger}_{\boldsymbol{\eta},2} & (k = 1), \\ g^{\dagger}_{\boldsymbol{\eta},1} & (k = 2), \\ g^{\dagger}_{\boldsymbol{\eta},k} & (\text{otherwise}). \end{cases}
$$

We now prove the first part by contradiction. Assume $g^{\dagger}_{\boldsymbol{\eta},1} > g^{\dagger}_{\boldsymbol{\eta},2}$. Then we have

$$
W_{\mathrm{APC}}(r_{\boldsymbol{\eta}}^{\dagger}, f'_{\boldsymbol{\eta}}; \boldsymbol{\eta}) - W_{\mathrm{APC}}(r_{\boldsymbol{\eta}}^{\dagger}, f_{\boldsymbol{\eta}}^{\dagger}; \boldsymbol{\eta})
$$

$$
= \sum_y \eta_y \left( \sum_{y' \neq y} \phi\big(\alpha(g'_{\boldsymbol{\eta},y} - g'_{\boldsymbol{\eta},y'} - r_{\boldsymbol{\eta}}^{\dagger})\big) \right) - \sum_y \eta_y \left( \sum_{y' \neq y} \phi\big(\alpha(g^{\dagger}_{\boldsymbol{\eta},y} - g^{\dagger}_{\boldsymbol{\eta},y'} - r_{\boldsymbol{\eta}}^{\dagger})\big) \right)
$$

$$
= \sum_{y=1,2} \eta_y \left( \sum_{y' \neq y} \phi\big(\alpha(g'_{\boldsymbol{\eta},y} - g'_{\boldsymbol{\eta},y'} - r_{\boldsymbol{\eta}}^{\dagger})\big) \right) - \sum_{y=1,2} \eta_y \left( \sum_{y' \neq y} \phi\big(\alpha(g^{\dagger}_{\boldsymbol{\eta},y} - g^{\dagger}_{\boldsymbol{\eta},y'} - r_{\boldsymbol{\eta}}^{\dagger})\big) \right)
$$

$$
= (\eta_2 - \eta_1) \Big[ \phi\big(\alpha(g^{\dagger}_{\boldsymbol{\eta},1} - g^{\dagger}_{\boldsymbol{\eta},2} - r_{\boldsymbol{\eta}}^{\dagger})\big) - \phi\big(\alpha(g^{\dagger}_{\boldsymbol{\eta},2} - g^{\dagger}_{\boldsymbol{\eta},1} - r_{\boldsymbol{\eta}}^{\dagger})\big)
$$

$$
+ \sum_{y'>2} \Big( \phi\big(\alpha(g^{\dagger}_{\boldsymbol{\eta},1} - g^{\dagger}_{\boldsymbol{\eta},y'} - r_{\boldsymbol{\eta}}^{\dagger})\big) - \phi\big(\alpha(g^{\dagger}_{\boldsymbol{\eta},2} - g^{\dagger}_{\boldsymbol{\eta},y'} - r_{\boldsymbol{\eta}}^{\dagger})\big) \Big) \Big]
$$

$$
< (\eta_2 - \eta_1)[0 + 0] = 0,
$$

which contradicts the optimality of $f_{\boldsymbol{\eta}}^{\dagger}$. Therefore we must have $g^{\dagger}_{\boldsymbol{\eta},1} \leq g^{\dagger}_{\boldsymbol{\eta},2}$, which proves the first part.

Next we assume $\phi$ is differentiable. In this case, the first-order optimality condition gives

$$
\eta_1 \sum_{y' \in \mathcal{Y}} \phi'\big(\alpha(g^{\dagger}_{\boldsymbol{\eta},1} - g^{\dagger}_{\boldsymbol{\eta},y'} - r_{\boldsymbol{\eta}}^{\dagger})\big) = \sum_{y' \in \mathcal{Y}} \eta_{y'} \phi'\big(\alpha(g^{\dagger}_{\boldsymbol{\eta},y'} - g^{\dagger}_{\boldsymbol{\eta},1} - r_{\boldsymbol{\eta}}^{\dagger})\big) \tag{38}
$$

$$
\eta_2 \sum_{y' \in \mathcal{Y}} \phi'\big(\alpha(g^{\dagger}_{\boldsymbol{\eta},2} - g^{\dagger}_{\boldsymbol{\eta},y'} - r_{\boldsymbol{\eta}}^{\dagger})\big) = \sum_{y' \in \mathcal{Y}} \eta_{y'} \phi'\big(\alpha(g^{\dagger}_{\boldsymbol{\eta},y'} - g^{\dagger}_{\boldsymbol{\eta},2} - r_{\boldsymbol{\eta}}^{\dagger})\big) \tag{39}
$$

by taking the derivative of $W_{\mathrm{APC}}(r, f; \boldsymbol{\eta})$ with respect to $g_1$ and $g_2$, and then substituting $(r_{\boldsymbol{\eta}}^{\dagger}, f_{\boldsymbol{\eta}}^{\dagger})$ for $(r, f)$. We again prove the second part by contradiction. Assume $g^{\dagger}_{\boldsymbol{\eta},1} = g^{\dagger}_{\boldsymbol{\eta},2} = g^{\dagger}_{\boldsymbol{\eta}}$. In this case, the RHSs of (38) and (39) are the same, which gives

$$
\eta_1 \sum_{y' \in \mathcal{Y}} \phi'\big(\alpha(g^{\dagger}_{\boldsymbol{\eta}} - g^{\dagger}_{\boldsymbol{\eta},y'} - r_{\boldsymbol{\eta}}^{\dagger})\big) = \eta_2 \sum_{y' \in \mathcal{Y}} \phi'\big(\alpha(g^{\dagger}_{\boldsymbol{\eta}} - g^{\dagger}_{\boldsymbol{\eta},y'} - r_{\boldsymbol{\eta}}^{\dagger})\big),
$$

or equivalently,

$$
(\eta_1 - \eta_2) \sum_{y' \in \mathcal{Y}} \phi'\big(\alpha(g^{\dagger}_{\boldsymbol{\eta}} - g^{\dagger}_{\boldsymbol{\eta},y'} - r_{\boldsymbol{\eta}}^{\dagger})\big) = 0. \tag{40}
$$

Note that $\sum_{y' \in \mathcal{Y}} \phi'\big(\alpha(g^{\dagger}_{\boldsymbol{\eta}} - g^{\dagger}_{\boldsymbol{\eta},y'} - r_{\boldsymbol{\eta}}^{\dagger})\big) \leq 2\phi'(-\alpha r_{\boldsymbol{\eta}}^{\dagger}) < 0$ holds since $\phi$ is a non-increasing function. Therefore we must have $\eta_1 = \eta_2$ for (40) to hold. However, this contradicts the assumption $\eta_1 < \eta_2$, therefore, we have $g^{\dagger}_{\boldsymbol{\eta},1} \neq g^{\dagger}_{\boldsymbol{\eta},2}$. Together with the fact of the first part, we have $g^{\dagger}_{\boldsymbol{\eta},1} < g^{\dagger}_{\boldsymbol{\eta},2}$ in this case. $\quad\square$

To compute $r_{\boldsymbol{\eta}}^{\dagger}$, we need true class distribution $\boldsymbol{\eta}$, which is unknown to the learner. Thus, it is difficult to verify the requirement $\phi(z - \alpha r_{\boldsymbol{\eta}}^{\dagger}) < \phi(-z - \alpha r_{\boldsymbol{\eta}}^{\dagger})$ $(z > 0)$ for $\mathcal{L}_{\mathrm{APC}}$. However, the following corollary, which immediately follows from Theorems 16 and 17, implies that logistic loss and exponential loss are good candidates for $\phi$ in $\mathcal{L}_{\mathrm{APC}}$ and $\mathcal{L}_{\mathrm{MPC}}$, respectively.

**Corollary 18** (Strictly order-preserving property for $\mathcal{L}_{\mathrm{APC}}, \mathcal{L}_{\mathrm{MPC}}$). *$\mathcal{L}_{\mathrm{APC}}$ and $\mathcal{L}_{\mathrm{MPC}}$ are strictly order-preserving if $\phi$ is a differentiable function such that $\phi'(z) < 0$ holds for all $z \in \mathbb{R}$.*

## B.4   Rejection calibration

In the following, we give a simple example to illustrate the intuition of Theorem 4 and Corollary 5. Throughout this section, we consider APC loss with exponential loss for $\phi$ and $\psi$.

$$\mathcal{L}_{\mathrm{APC}}(r, f; \boldsymbol{x}, y) = \sum_{y' \neq y} \exp\Big(\alpha\big(r(\boldsymbol{x}) + g_{y'}(\boldsymbol{x}) - g_y(\boldsymbol{x})\big)\Big) + c\exp\big(-\beta r(\boldsymbol{x})\big),$$

$$W_{\mathrm{APC}}(r, f; \boldsymbol{\eta}) = \sum_y \eta_y \left(\sum_{y' \neq y} \exp\Big(\alpha\big(r + g_{y'} - g_y\big)\Big)\right) + c\exp\big(-\beta r\big).$$

Note that $\mathcal{L}_{\mathrm{MPC}} = \mathcal{L}_{\mathrm{APC}}$ when we use exponential loss for binary losses.

**Binary Case**   In the binary case, $\mathcal{L}_{\mathrm{APC}}$ and $W_{\mathrm{APC}}$ are expressed as

$$\mathcal{L}_{\mathrm{APC}}(r, f; \boldsymbol{x}, y) = \exp\big[\alpha(r(\boldsymbol{x}) - yf(\boldsymbol{x}))\big] + c\exp\big[-\beta r(\boldsymbol{x})\big],$$
$$W_{\mathrm{APC}}(r, f; \boldsymbol{\eta}) = \eta_+ \exp\big[\alpha(r - f)\big] + \eta_- \exp\big[\alpha(r + f)\big] + c\exp(-\beta r),$$

which coincide with the losses defined in Cortes et al. [9]. Since $f_{\boldsymbol{\eta}}^{\dagger}$ is the minimizer of $W_{\mathrm{APC}}$, by taking the derivative of $W_{\mathrm{APC}}$ with respect to $f$ and setting it to zero, we get $f_{\boldsymbol{\eta}}^{\dagger} = \frac{1}{2\alpha}\log\frac{\eta_+}{\eta_-}$. Thus, $\frac{\partial W_{\mathrm{APC}}(r, f_{\boldsymbol{\eta}}^{\dagger}; \boldsymbol{\eta})}{\partial r}$ can be expressed as follows:

$$\frac{\partial W_{\mathrm{APC}}(r, f_{\boldsymbol{\eta}}^{\dagger}; \boldsymbol{\eta})}{\partial r} = \alpha\eta_+ \exp\big[\alpha(r - f_{\boldsymbol{\eta}}^{\dagger})\big] + \alpha\eta_- \exp\big[\alpha(r + f_{\boldsymbol{\eta}}^{\dagger})\big] - c\beta\exp(-\beta r)$$
$$= 2\alpha\sqrt{\eta_+ \eta_-}\exp(\alpha r) - c\beta\exp(-\beta r).$$

Hence, we have

$$\sup_{\max_y \eta_y \geq 1-c} \left.\frac{\partial W_{\mathrm{APC}}(r, f_{\boldsymbol{\eta}}^{\dagger}; \boldsymbol{\eta})}{\partial r}\right|_{r=0} = \sup_{\max_y \eta_y = 1-c} 2\alpha\sqrt{\eta_+\eta_-} - c\beta = 2\alpha\sqrt{c(1-c)} - c\beta,$$

$$\inf_{\max_y \eta_y = 1-c} \left.\frac{\partial W_{\mathrm{APC}}(r, f_{\boldsymbol{\eta}}^{\dagger}; \boldsymbol{\eta})}{\partial r}\right|_{r=0} = \inf_{\max_y \eta_y = 1-c} 2\alpha\sqrt{\eta_+\eta_-} - c\beta = 2\alpha\sqrt{c(1-c)} - c\beta.$$

Using the result of Theorem 4, we can confirm that rejection calibration holds if and only if

$$2\alpha\sqrt{c(1-c)} - c\beta = 0 \quad \Leftrightarrow \quad \frac{\beta}{\alpha} = 2\sqrt{\frac{1-c}{c}}, \tag{41}$$

which coincides with the result of Theorem 1 of Cortes et al. [9]. This suggests that Theorem 4 is a general extension of their result.

**Multiclass case**   Next, we consider the multiclass case, i.e., the case where $K > 2$. We assume that $c < \frac{1}{2}$, otherwise, even data points with low confidence will also be accepted. Since $\boldsymbol{g}_{\boldsymbol{\eta}}^{\dagger}$ is the minimizer of $W_{\mathrm{APC}}$, by taking the derivative of $W_{\mathrm{APC}}$ with respect to $\boldsymbol{g}$ and setting it zero, we

get $g^{\dagger}_{\boldsymbol{\eta},y} - g^{\dagger}_{\boldsymbol{\eta},y'} = \frac{1}{2\alpha} \log \frac{\eta_y}{\eta_{y'}}$. Thus, $\frac{\partial W_{\mathrm{APC}}(r, f^{\dagger}_{\boldsymbol{\eta}}; \boldsymbol{\eta})}{\partial r}$ can be calculated as follows:

$$
\begin{aligned}
\frac{\partial W_{\mathrm{APC}}(r, f^{\dagger}_{\boldsymbol{\eta}}; \boldsymbol{\eta})}{\partial r} &= \alpha \exp(\alpha r) \sum_y \eta_y \sum_{y \neq y'} \exp\left[\alpha(g^{\dagger}_{\boldsymbol{\eta},y'} - g^{\dagger}_{\boldsymbol{\eta},y})\right] - c\beta \exp(-\beta r) \\
&= \alpha \exp(\alpha r) \sum_y \sum_{y' \neq y} \sqrt{\eta_y \eta_{y'}} - c\beta \exp(-\beta r) \\
&= \alpha \exp(\alpha r) \sum_y \left( \sum_{y'} \sqrt{\eta_y \eta_{y'}} - \eta_y \right) - c\beta \exp(-\beta r) \\
&= \alpha \exp(\alpha r) \left( \left( \sum_y \sqrt{\eta_y} \right)^2 - 1 \right) - c\beta \exp(-\beta r),
\end{aligned}
$$

where in the last line we used the condition $\sum_y \eta_y = 1$. We next see how Eqs. (6) (left) and (6) (right) behave. As for the Eq. (6) (left), we have

$$
\begin{aligned}
\sup_{\max_y \eta_y = 1-c} \frac{\partial W_{\mathrm{APC}}(r, f^{\dagger}_{\boldsymbol{\eta}}; \boldsymbol{\eta})}{\partial r}\bigg|_{r=0} &= \sup_{\max_y \eta_y = 1-c} \alpha \left( \left( \sum_y \sqrt{\eta_y} \right)^2 - 1 \right) - c\beta \\
&= \alpha \left( \left( \sqrt{1-c} + \sqrt{\frac{c}{K-1}}(K-1) \right)^2 - 1 \right) - c\beta \\
&= \alpha \left( (K-2)c + 2\sqrt{(K-1)c(1-c)} \right) - c\beta.
\end{aligned}
$$

Note that since $c < \frac{1}{2}$, the supremum is satisfied when $\max_y \eta_y = 1-c$, and $\eta_{y'} = \frac{c}{K-1}$ for the others. The above calculation gives the condition

$$
\frac{\beta}{\alpha} \geq (K-2) + 2\sqrt{(K-1)\frac{1-c}{c}}. \tag{42}
$$

When $K = 2$, the RHS of (42) is the same as RHS of (41). As for Eq. (6) (right), we get

$$
\begin{aligned}
\inf_{\max_y \eta_y \leq 1-c} \frac{\partial W_{\mathrm{APC}}(0, f^{\dagger}; \boldsymbol{\eta})}{\partial r} &= \inf_{\max_y \eta_y \leq 1-c} \alpha \left( \left( \sum_y \sqrt{\eta_y} \right)^2 - 1 \right) - c\beta \\
&= 2\alpha\sqrt{c(1-c)} - c\beta.
\end{aligned}
$$

Note that since $c < \frac{1}{2}$, the infimum is satisfied when $\max_y \eta_y = 1-c$, and $\eta_{y'} = c$, and $\eta_{y''} = 0$ for the others. The above calculation gives the condition:

$$
\frac{\beta}{\alpha} \leq 2\sqrt{\frac{1-c}{c}}. \tag{43}
$$

Again, when $K = 2$, the RHS of (43) is the same as RHS of (41).

However, when we deal with multiclass classification, we can easily confirm that (42) and (43) cannot simultaneously be satisfied, since

$$
(K-2) + 2\sqrt{(K-1)\frac{1-c}{c}} > 2\sqrt{\frac{1-c}{c}}.
$$

The intuition of this result is that we cannot achieve rejection calibration in multiclass setting, using classifier-rejector approach. More precisely, if we set hyper-parameters $\alpha$ and $\beta$ to satisfy (42), we can make FR to zero, but we cannot make FA to zero. Conversely, if we set hyper-parameters $\alpha$ and $\beta$ to satisfy (43), we can make FA to zero, but we cannot make FR to zero.

For the logistic loss, we get $g^{\dagger}_{\boldsymbol{\eta},y} - g^{\dagger}_{\boldsymbol{\eta},y'} = \frac{1}{\alpha} \log \frac{\eta_y}{\eta_{y'}}$. After applying the same procedure as we did for the proof of the exponential loss, the failure result of the logistic loss can be obtained.

# C Experiment details

## C.1 Synthetic datasets

- Goal: To see the calibration result of proposed method.
- Datasets:
    - We randomly select 8 two-dimensional vectors $\boldsymbol{\mu}_1, \ldots, \boldsymbol{\mu}_8 \in \mathbb{R}^2$. These 8 vectors correspond to 8 classes.
    - Each sample $(\boldsymbol{x}, y)$ is sampled from $p(y)p(\boldsymbol{x}|y)$, where $p(y)$ is a uniform distribution $p(y) = \frac{1}{8}$, and $p(\boldsymbol{x}|y)$ is a Gaussian distribution $\mathcal{N}(\boldsymbol{\mu}_y, 0.2I_2)$. Here, $I_2$ is a $2 \times 2$ identity matrix.
    - (# training data): $\{20, 50, 100, 200, 500, 1000, 1500, 2000, 5000, 10000\}$ for each class.
- Rejection Cost: $c \in \{0.05, 0.1, 0.2, 0.3, 0.4\}$.
- Methods:
    - APC loss (8) with logistic loss and exponential loss. We set $\alpha = 1$ for simplicity and $\beta$ is set to satisfy (6) (left) and (6) (right) respectively (APC+log+acc, APC+log+rej, APC+exp+acc, APC+exp+rej).
    - MPC loss (7) with logistic loss (MPC+log). Note that MPC loss with exponential loss reduces to APC+exp. We set $\alpha = 1$ for simplicity and $\beta$ is set to satisfy (6) (left) and (6) (right) respectively (MPC+log+acc, MPC+log+rej).
    - OVA loss with logistic loss and exponential loss (OVA+log, OVA+exp)
    - CE loss (CE)
- Hyper-parameter Selection:
    - $\ell_2$ regularization, where weight decays are chosen from $\{10^{-7}, 10^{-4}, 10^{-1}\}$.
    - We did 80-20 split for each training data for validation for hyper-parameter tuning.
    - Using a different random partition, we repeated the experiments three times.
- Optimization:
    - AMSGRAD with 100 epochs.
- Model:
    - one-hidden-layer neural network (d-3-1) with rectified linear units (ReLU) as activation functions.

## C.2 Benchmark datasets

- Goal: To see the empirical performance including the existing method.
- Datasets: see Table 3. They can all be downloaded from https://archive.ics.uci.edu/ml/ or https://www.csie.ntu.edu.tw/~cjlin/libsvmtools/datasets/multiclass.html.
- Rejection Cost: $c \in \{0.05, 0.1, 0.2, 0.3, 0.4\}$.
- Methods:
    - APC loss (8) with logistic loss and exponential loss (APC+log, APC+exp).
    - MPC loss (7) with logistic loss (MPC+log). Note that MPC loss with exponential loss reduces to APC+exp.
    - OVA loss with logistic loss and exponential loss (OVA+log, OVA+exp)
    - CE loss (CE)
    - existing method in Ramaswamy et al. [20] (OVA+hin)
- Hyper-parameter Selection:
    - $\ell_2$ regularization, where weight decays are chosen from $\{10^{-7}, 10^{-4}, 10^{-1}\}$.
    - For APC+log, APC+exp, MPC+log, we need to decide the parameter $\alpha$ and $\beta$. We set $\alpha = 1$. For $\beta$, three candidates are chosen from (6) (left), (6) (right) and the mean value of these values.

- For OVA+hin, five candidates of threshold parameter are chosen from $\{-0.95, -0.5, 0, 0.5, 0.95\}$.
- We did 80-20 split for each training data for validation for hyper-parameter tuning.
- Using a different random partition, we repeated the experiments ten times.

- Optimization:
  - AMSGRAD with 150 epochs.

- Model:
  - one-hidden-layer neural network (d-50-1) with rectified linear units (ReLU) as activation functions.

Figure 4: Average $0$-$1$-$c$ risk on the test set as a function of the rejection cost on benchmark datasets (full version).

Table 6: Mean and standard deviation of $0$-$1$-$c$ risks for 10 trials. Best and equivalent methods (with 5% t-test) are shown in bold face.

| dataset | $c$ | APC+log | APC+exp | MPC+log | OVA+log | OVA+exp | CE | OVA+hin |
|---|---|---|---|---|---|---|---|---|
| vehicle | 0.05 | 0.045 (0.0023) | **0.038** (**0.0081**) | 0.044 (0.0101) | 0.043 (0.0008) | **0.035** (**0.0030**) | **0.036** (**0.0007**) | 0.042 (0.0049) |
| | 0.1 | 0.074 (0.0038) | **0.073** (**0.0171**) | 0.07 (0.0092) | 0.074 (0.0036) | **0.064** (**0.0033**) | **0.063** (**0.0032**) | 0.085 (0.0060) |
| | 0.2 | 0.12 (0.0044) | 0.117 (0.0108) | 0.125 (0.0142) | 0.117 (0.0039) | **0.108** (**0.0037**) | **0.110** (**0.0020**) | 0.147 (0.0089) |
| | 0.3 | 0.157 (0.0068) | 0.157 (0.0076) | 0.163 (0.0150) | 0.156 (0.0058) | **0.152** (**0.0106**) | **0.148** (**0.0046**) | 0.184 (0.0088) |
| | 0.4 | **0.182** (**0.0130**) | 0.205 (0.0182) | 0.195 (0.0118) | 0.205 (0.0092) | 0.193 (0.0058) | **0.182** (**0.0057**) | 0.211 (0.0073) |
| satimage | 0.05 | **0.030** (**0.0013**) | 0.039 (0.0052) | 0.039 (0.0070) | **0.030** (**0.0011**) | **0.030** (**0.0011**) | **0.030** (**0.0006**) | 0.032 (0.0003) |
| | 0.1 | 0.052 (0.0013) | 0.057 (0.0041) | 0.063 (0.0069) | **0.049** (**0.0014**) | 0.050 (0.0008) | **0.049** (**0.0009**) | 0.057 (0.002) |
| | 0.2 | 0.087 (0.0027) | 0.093 (0.0055) | 0.094 (0.0048) | 0.081 (0.0016) | 0.081 (0.0029) | **0.078** (**0.0009**) | 0.080 (0.0012) |
| | 0.3 | 0.104 (0.0037) | 0.111 (0.0048) | 0.112 (0.0033) | 0.102 (0.0026) | 0.102 (0.0028) | **0.097** (**0.0013**) | 0.103 (0.0010) |
| | 0.4 | 0.115 (0.0036) | 0.113 (0.0030) | 0.116 (0.0026) | 0.114 (0.0036) | 0.116 (0.0033) | **0.107** (**0.0025**) | 0.122 (0.0019) |
| yeast | 0.05 | 0.050 (0.0000) | 0.057 (0.0109) | 0.052 (0.0023) | 0.050 (0.0000) | 0.051 (0.0009) | 0.050 (0.0000) | **0.050** (**0.0002**) |
| | 0.1 | **0.100** (**0.0000**) | 0.104 (0.0071) | 0.102 (0.0035) | **0.100** (**0.0006**) | 0.102 (0.0011) | **0.100** (**0.0006**) | **0.100** (**0.0002**) |
| | 0.2 | **0.200** (**0.0000**) | 0.222 (0.0297) | **0.200** (**0.0001**) | 0.201 (0.0009) | **0.201** (**0.0023**) | **0.200** (**0.0013**) | **0.200** (**0.0007**) |
| | 0.3 | 0.300 (0.0000) | 0.317 (0.0214) | 0.299 (0.0009) | 0.297 (0.0020) | 0.298 (0.0033) | **0.292** (**0.0020**) | 0.295 (0.0036) |
| | 0.4 | 0.400 (0.0009) | 0.410 (0.0104) | 0.412 (0.0117) | 0.388 (0.0031) | 0.395 (0.0050) | **0.374** (**0.0029**) | **0.372** (**0.0046**) |
| covtype | 0.05 | 0.052 (0.0007) | 0.057 (0.0016) | 0.059 (0.0012) | 0.055 (0.0012) | 0.056 (0.0015) | 0.056 (0.0018) | **0.050** (**0.0001**) |
| | 0.1 | 0.107 (0.0014) | 0.112 (0.0046) | 0.114 (0.0034) | 0.110 (0.0019) | 0.114 (0.0035) | 0.111 (0.0034) | **0.102** (**0.0005**) |
| | 0.2 | 0.211 (0.0039) | 0.210 (0.0059) | 0.216 (0.0061) | 0.210 (0.0028) | 0.216 (0.0070) | 0.208 (0.0064) | **0.196** (**0.0011**) |
| | 0.3 | 0.295 (0.0024) | 0.287 (0.0046) | 0.292 (0.0041) | 0.293 (0.0046) | 0.300 (0.0090) | **0.285** (**0.0090**) | **0.287** (**0.0015**) |
| | 0.4 | 0.349 (0.0047) | 0.364 (0.0123) | 0.366 (0.0147) | 0.353 (0.0063) | 0.360 (0.0113) | **0.339** (**0.0117**) | 0.373 (0.002) |
| letter | 0.05 | 0.040 (0.0013) | 0.038 (0.0015) | 0.033 (0.0013) | 0.041 (0.0007) | 0.036 (0.0010) | **0.032** (**0.0008**) | 0.041 (0.0007) |
| | 0.1 | 0.067 (0.0024) | 0.066 (0.0019) | 0.057 (0.0028) | 0.071 (0.0011) | 0.064 (0.0015) | **0.054** (**0.0019**) | 0.080 (0.0018) |
| | 0.2 | 0.103 (0.0035) | 0.109 (0.0025) | 0.093 (0.0045) | 0.118 (0.0018) | 0.110 (0.0028) | **0.083** (**0.0018**) | 0.146 (0.0046) |
| | 0.3 | 0.131 (0.0094) | 0.148 (0.0064) | 0.121 (0.0041) | 0.154 (0.0024) | 0.143 (0.0032) | **0.105** (**0.0016**) | 0.191 (0.0078) |
| | 0.4 | 0.149 (0.0080) | 0.166 (0.0076) | 0.148 (0.0097) | 0.179 (0.0033) | 0.168 (0.0036) | **0.120** (**0.0021**) | 0.214 (0.0094) |

Table 7: Mean and standard deviation of accuracy on non rejected data for 10 trials. "-" corresponds to the case where all the test data samples are rejected.

| dataset | $c$ | APC+log | APC+exp | MPC+log | OVA+log | OVA+exp | CE | OVA+hin |
|---|---|---|---|---|---|---|---|---|
| vehicle | 0.05 | - ( - ) | 0.981 (0.0204) | 0.966 (0.0231) | 1.000 (0.0000) | 0.991 (0.0089) | 1.000 (0.0000) | 0.996 (0.0111) |
| | 0.1 | 1.000 (0.0000) | 0.962 (0.0348) | 0.958 (0.0209) | 0.989 (0.0115) | 0.981 (0.0064) | 0.990 (0.0081) | 0.947 (0.0282) |
| | 0.2 | 0.984 (0.0188) | 0.937 (0.0289) | 0.924 (0.0301) | 0.979 (0.0072) | 0.972 (0.0044) | 0.974 (0.0005) | 0.964 (0.0535) |
| | 0.3 | 0.946 (0.0250) | 0.905 (0.0195) | 0.894 (0.0384) | 0.960 (0.0093) | 0.945 (0.0198) | 0.959 (0.0073) | 0.941 (0.0265) |
| | 0.4 | 0.891 (0.0288) | 0.831 (0.0543) | 0.853 (0.0418) | 0.902 (0.0164) | 0.904 (0.0107) | 0.917 (0.0087) | 0.887 (0.0375) |
| satimage | 0.05 | 0.991 (0.0025) | 0.973 (0.0148) | 0.972 (0.0141) | 0.987 (0.0014) | 0.982 (0.0020) | 0.983 (0.0011) | 0.995 (0.0010) |
| | 0.1 | 0.975 (0.0034) | 0.966 (0.0095) | 0.957 (0.0170) | 0.980 (0.0023) | 0.973 (0.0010) | 0.975 (0.0015) | 0.974 (0.0093) |
| | 0.2 | 0.950 (0.0102) | 0.930 (0.0117) | 0.926 (0.0119) | 0.962 (0.0022) | 0.954 (0.0043) | 0.957 (0.0013) | 0.965 (0.0018) |
| | 0.3 | 0.929 (0.0055) | 0.904 (0.0196) | 0.905 (0.0153) | 0.944 (0.0031) | 0.935 (0.0023) | 0.938 (0.0020) | 0.952 (0.0013) |
| | 0.4 | 0.915 (0.0066) | 0.890 (0.0049) | 0.890 (0.0109) | 0.922 (0.0034) | 0.915 (0.0029) | 0.918 (0.0023) | 0.933 (0.0028) |
| yeast | 0.05 | - ( - ) | - ( - ) | - ( - ) | - ( - ) | - ( - ) | - ( - ) | - ( - ) |
| | 0.1 | - ( - ) | - ( - ) | - ( - ) | - ( - ) | 0.593 (0.2699) | - ( - ) | - ( - ) |
| | 0.2 | - ( - ) | - ( - ) | - ( - ) | - ( - ) | 0.742 (0.1538) | 0.806 (0.0615) | - ( - ) |
| | 0.3 | - ( - ) | - ( - ) | - ( - ) | 0.822 (0.0780) | 0.733 (0.0577) | 0.805 (0.0301) | - ( - ) |
| | 0.4 | - ( - ) | - ( - ) | - ( - ) | 0.750 (0.0393) | 0.630 (0.0353) | 0.766 (0.0169) | 0.760 (0.0394) |
| covtype | 0.05 | 0.795 (0.0205) | 0.797 (0.0764) | 0.798 (0.0169) | 0.821 (0.0267) | 0.803 (0.0313) | 0.820 (0.0321) | 0.886 (0.0314) |
| | 0.1 | 0.765 (0.0176) | 0.806 (0.0177) | 0.793 (0.0185) | 0.781 (0.0200) | 0.767 (0.0375) | 0.796 (0.0285) | 0.812 (0.0216) |
| | 0.2 | 0.740 (0.0181) | 0.759 (0.0185) | 0.738 (0.0102) | 0.749 (0.0135) | 0.732 (0.0317) | 0.771 (0.0235) | 0.850 (0.0161) |
| | 0.3 | 0.719 (0.0115) | 0.743 (0.0156) | 0.722 (0.0116) | 0.719 (0.0126) | 0.699 (0.0231) | 0.733 (0.0204) | 0.795 (0.0133) |
| | 0.4 | 0.698 (0.0134) | 0.638 (0.0128) | 0.649 (0.0340) | 0.687 (0.0109) | 0.669 (0.0183) | 0.694 (0.0180) | 0.768 (0.0135) |
| letter | 0.05 | 0.998 (0.0010) | 0.986 (0.0046) | 0.986 (0.0021) | 0.996 (0.0015) | 0.994 (0.0019) | 0.998 (0.0008) | 0.996 (0.0021) |
| | 0.1 | 0.993 (0.0013) | 0.978 (0.0045) | 0.980 (0.0034) | 0.994 (0.0014) | 0.986 (0.0019) | 0.994 (0.0015) | 0.963 (0.0044) |
| | 0.2 | 0.979 (0.0027) | 0.966 (0.0049) | 0.969 (0.0046) | 0.983 (0.0015) | 0.968 (0.0019) | 0.984 (0.0014) | 0.913 (0.0227) |
| | 0.3 | 0.969 (0.0054) | 0.948 (0.0355) | 0.961 (0.0038) | 0.966 (0.0025) | 0.950 (0.0023) | 0.969 (0.0024) | 0.942 (0.0446) |
| | 0.4 | 0.952 (0.0051) | 0.852 (0.0379) | 0.946 (0.0383) | 0.946 (0.0023) | 0.930 (0.0027) | 0.949 (0.0026) | 0.892 (0.0484) |

Table 8: Mean and standard deviation of rejection ratio (the ratio of rejected data samples over whole test data) for 10 trials.

| dataset | $c$ | APC+log | APC+exp | MPC+log | OVA+log | OVA+exp | CE | OVA+hin |
|---------|-----|---------|---------|---------|---------|---------|-----|---------|
| vehicle | 0.05 | 0.909 (0.0460) | 0.605 (0.0486) | 0.570 (0.0400) | 0.868 (0.0156) | 0.623 (0.0179) | 0.721 (0.0137) | 0.825 (0.0914) |
| | 0.1 | 0.740 (0.0383) | 0.525 (0.0812) | 0.469 (0.0465) | 0.708 (0.0271) | 0.556 (0.0129) | 0.590 (0.0105) | 0.567 (0.1859) |
| | 0.2 | 0.564 (0.0365) | 0.381 (0.0842) | 0.374 (0.0852) | 0.538 (0.0131) | 0.466 (0.0224) | 0.483 (0.0098) | 0.620 (0.1821) |
| | 0.3 | 0.410 (0.0633) | 0.299 (0.0616) | 0.276 (0.0841) | 0.447 (0.0080) | 0.396 (0.0200) | 0.412 (0.0149) | 0.512 (0.0535) |
| | 0.4 | 0.247 (0.0450) | 0.123 (0.1347) | 0.173 (0.1082) | 0.353 (0.0127) | 0.321 (0.0126) | 0.313 (0.0092) | 0.332 (0.0839) |
| satimage | 0.05 | 0.504 (0.0359) | 0.428 (0.1361) | 0.411 (0.1312) | 0.462 (0.0125) | 0.385 (0.0105) | 0.400 (0.0125) | 0.603 (0.0060) |
| | 0.1 | 0.360 (0.0245) | 0.347 (0.0466) | 0.314 (0.0981) | 0.357 (0.0050) | 0.311 (0.0065) | 0.313 (0.0082) | 0.409 (0.0781) |
| | 0.2 | 0.245 (0.0407) | 0.173 (0.0432) | 0.158 (0.0560) | 0.264 (0.0032) | 0.228 (0.0094) | 0.221 (0.0075) | 0.275 (0.0042) |
| | 0.3 | 0.142 (0.0247) | 0.069 (0.0679) | 0.074 (0.0609) | 0.187 (0.0050) | 0.155 (0.0077) | 0.145 (0.0079) | 0.219 (0.0037) |
| | 0.4 | 0.094 (0.0221) | 0.011 (0.0090) | 0.020 (0.0301) | 0.113 (0.0030) | 0.096 (0.0051) | 0.078 (0.0042) | 0.166 (0.0066) |
| yeast | 0.05 | 1.000 (0.0000) | 0.970 (0.0441) | 0.985 (0.0134) | 1.000 (0.0000) | 0.999 (0.0010) | 1.000 (0.0000) | 0.998 (0.0036) |
| | 0.1 | 1.000 (0.0000) | 0.971 (0.0265) | 0.982 (0.0224) | 0.999 (0.0010) | 0.995 (0.0026) | 0.999 (0.0019) | 0.999 (0.0025) |
| | 0.2 | 1.000 (0.0000) | 0.879 (0.1466) | 0.999 (0.0025) | 0.992 (0.0039) | 0.977 (0.0062) | 0.979 (0.0055) | 0.994 (0.0116) |
| | 0.3 | 1.000 (0.0000) | 0.858 (0.1741) | 0.996 (0.004) | 0.974 (0.0088) | 0.931 (0.0105) | 0.918 (0.0068) | 0.950 (0.0271) |
| | 0.4 | 0.998 (0.0068) | 0.581 (0.3448) | 0.593 (0.3394) | 0.919 (0.0135) | 0.843 (0.0197) | 0.845 (0.0158) | 0.816 (0.0434) |
| covtype | 0.05 | 0.985 (0.0025) | 0.950 (0.0194) | 0.943 (0.0102) | 0.964 (0.0037) | 0.957 (0.0059) | 0.955 (0.0051) | 0.995 (0.0009) |
| | 0.1 | 0.947 (0.0104) | 0.877 (0.0304) | 0.866 (0.0188) | 0.913 (0.0062) | 0.895 (0.0132) | 0.892 (0.0073) | 0.982 (0.0019) |
| | 0.2 | 0.816 (0.0293) | 0.762 (0.0426) | 0.736 (0.0747) | 0.793 (0.0098) | 0.759 (0.0238) | 0.733 (0.0136) | 0.924 (0.0072) |
| | 0.3 | 0.671 (0.0657) | 0.688 (0.0328) | 0.614 (0.0562) | 0.641 (0.0141) | 0.600 (0.0344) | 0.553 (0.0141) | 0.860 (0.007) |
| | 0.4 | 0.470 (0.0647) | 0.031 (0.0091) | 0.156 (0.2154) | 0.457 (0.0163) | 0.421 (0.0391) | 0.346 (0.0120) | 0.839 (0.0072) |
| letter | 0.05 | 0.792 (0.0265) | 0.660 (0.0245) | 0.538 (0.0333) | 0.802 (0.0107) | 0.682 (0.0119) | 0.628 (0.0180) | 0.810 (0.0192) |
| | 0.1 | 0.646 (0.0283) | 0.571 (0.0291) | 0.460 (0.0475) | 0.695 (0.0103) | 0.585 (0.0130) | 0.506 (0.0164) | 0.677 (0.0210) |
| | 0.2 | 0.461 (0.0251) | 0.451 (0.0205) | 0.366 (0.0387) | 0.552 (0.0076) | 0.463 (0.0123) | 0.365 (0.0123) | 0.507 (0.0804) |
| | 0.3 | 0.371 (0.0444) | 0.369 (0.1188) | 0.316 (0.0218) | 0.451 (0.0063) | 0.372 (0.0101) | 0.273 (0.0091) | 0.528 (0.1163) |
| | 0.4 | 0.286 (0.0280) | 0.055 (0.1173) | 0.262 (0.0785) | 0.363 (0.0075) | 0.295 (0.0112) | 0.198 (0.0078) | 0.346 (0.1134) |