[Reviews · NeurIPS 2019]

Reviewer 1



Since 2017, there has been a considerable effort in improving confidence modeling with classifiers, with 2 majors goals: rejection when uncertain, and detecting out-of-distribution examples. In a work that has been mostly empirical and focused on DNNs, this line of work stands out by being mostly theoretical, taking its seeds from work with boosting with abstention. There seems two main contributions in this work, using excellent theoretical derivations. However, their significance may be limited as the authors do not make any effort to connect them to the deep learning literature: 1) Negative result: In some multiclass setting using rejection, it is pointless to train a separate rejector. Solutions that converges towards the Bayes optimal solution requires the rejector to be a function of the Bayes-optimal scoring function, that is it should not be trained separately. 2) New Bound: The excess bound loss for CE (theorem 8) which clearly states that one can train with the cross-entropy loss (usual loss for Softmax DNNs) without taking the rejection threshold 'C' into account. This is a very nice result, which confirms theoretically what most people were already doing empirically. I am not sure is this is what the authors mean by "novel rejection criteria" in the abstract, but the main lessons I am taking from this paper is that they confirm me in what I was doing as a DNN practitioner. This paper would greatly gain in significance with better connection to the Deep Learning literature. 1) The scope of the negative result is unclear. The example given is over over losses (MPC/APC with binary exponential loss) I have never seen used and can only imagine in the boosting literature. There has been recent attempts to train separate rejectors in DNN settings. See for instance "Learning Confidence for Out-of-Distribution Detection in Neural Networks" (https://arxiv.org/pdf/1802.04865.pdf), where the authors spend enormous effort taming a very unstable training procedure over the loss in their Eq.(5) with hacks, in particular in determining a hyper parameter lambda. Could we apply the conditions expressed in Eq.(6) to their work? 2) The new bound that justifies training using Softmax and cross-entropy should be better publicized. Which leaves us facing the same mystery: why does SGD fail so badly to converge towards the Bayes-optimal, producing over-confident outputs (see https://arxiv.org/abs/1706.04599 "on calibration of modern neural network"). One key part of the paper which would greatly benefit from more intuitive explanations is Section 3. The results are first presented without justification and I could not understand the explanations given from line 144-158. While the paper is well written, the English could be improved and some explanation are sometimes unclear or confusing. A few detailed comments by line: 74 "It is well known": The Bayes-optimal rejector is not trivial, and some do not agree with it. It should be traced to Chow. The fact that the threshold should be (1-c) is not 'well known'. 128 typo "seperation" 145 Is the "objective function" W or dW/dr ? 171 MPS -> MPC 174 State that MPC and APC are the same with exp loss! Actually, this part is very hard to read: the distinction between MPC and APC does not add anything to the understanding of the paper and could be moved to the appendix. 227 Do you mean? This enables us to derive the same bound in a considerably simpleR way This enables us to derive a more general bound in a simple way 277 "can no more" ... "unlike" is a strange construction.

Reviewer 2



After Rebuttal: The clarification of the negative result is still not sufficient. The authors keep saying "difficult" what does that mean? If theorem 4 is only a condition for checking calibrated surrogates, it actually reduces in value. The authors should make a serious effort into making a mathematically coherent impossibility result and not just state this is "difficult". I also disagree that the excess risk derivations are complicated. Simple application of Pinsker's inequality will transform the excess log loss into a l1 norm distance, which can be easily converted to a excess abstain loss risk. Before Rebuttal: Summary: The authors consider the problem of multiclass classification with a reject option. Assuming the "cost of abstaining" is a constant, the authors analyze various surrogates and determine whether they are "calibrated" to the abstain loss. There are two main paradigms for building abstaining classifier : 1. confidence based: build a scoring model, and abstain if the scores are not "confident" 2. separation based: build a separate rejector and scorer, and use the scorer to classify all instances that have not been rejected by the rejector. The authors argue that the separation based methods cannot be calibrated, and showstandard confidence based methods can be made calibrated by use of an appropriate threshold for rejection. Review: The main contribution of the paper would be an attempt at showing the impossibility of "separation-based" calibrated surrogates. (Theorem 4). Theorem 4 looks correct and indicates some problems with a surrogate that has to be calibrated with the abstain loss, but there is no concrete impossibility statement. Lines 150 to 155, try to make this precise but it is not particularly clear. This should be made into a theorem or a corollary. I am guessing it should be something along the lines of "if the surrogate is convex in r, then it is not calibrated w.r.t the abstain loss". If this is what the authors mean, they should state and prove it instead of asking the readers to look at a statement regarding the derivatives and intuit it. The section on confidence based surrogates is standard, and cannot be considered an original contribution. It is well known that with a proper multiclass loss, the class probabilities can be estimated, and the form of the confidence predictor is exactly the same as the form of the Bayes classifier. There are some excess risk bounds derived in theorem 7 and Table 1, but these are not particularly original and such bounds can be derived using previously known techniques , (a la Steinwart, Bartlett et al.) The empirical results are not particularly impressive (or original, as the algorithm proposed is simply standard multinomial logistic regression). The APC,MPC methods have been argued to be sub-optimal in theory and are shown sub-optimal in practice, which is just a little bit satisfying.

Reviewer 3



This paper studies the problem of multiclass classification with rejection. The authors firstly survey the confidence-based approach and separation-based approach based on binary classification, and then extend these two kinds of approaches to multi-class cases. For separation-based approach, the authors defined a necessary condition for rejection calibration. For confidence-based approach, they discussed the one-versus-all loss and cross-entropy loss for multi-class cases. The error boundaries of related losses were also analyzed. Some experiments were also implemented. Although this is not the first paper to discuss the problem of mutli-class classification with rejection, the contribution of this paper is that they provide many theoretical analysis and experimental comparison for the related problem, including some theorems and some interesting experimental conclusions. There are many theorems and proofs in Appendix, I just checked two of them and they are all correct. The experiments appeared in the paper and Appendix are also convinced. I think this paper is useful for the research area of learning with rejection.

[Author Response · NeurIPS 2019]

Thank you for helpful comments and suggestions. We will address the concerns raised by the reviewers.

Response to Reviewer#1

(Q1) A scope of negative result is unclear.
(A1) The negative result of the separation-based approach comes from Corollary 5, which indicates a necessary
condition for a surrogate loss of the $0$-$1$-$c$ loss to be rejection calibrated. In our paper, we demonstrate that if a surrogate
loss is MPC or APC using the exponential loss, it will not satisfy this condition and therefore it is not calibrated. Note
that our negative result is not limited to our demonstration. If one designs a new surrogate loss, one may use this
corollary to verify whether that surrogate loss satisfies this necessary condition and may further check if it is sufficient
for rejection calibration with Theorem 4.

We discuss the difficulty to satisfy Corollary 5 when the problem becomes multiclass in Lines 144–158. In Corollary 5,
the condition under the supremum and infimum operators is identical, which is $\boldsymbol{\eta}\colon \max_y \eta_y = 1 - c$. In the binary
case, if $\max_y \eta_y = 0.7$, then the class probability of the other class is uniquely determined as $0.3$, since they must sum
to one. This makes it easier to tune the hyperparameters to satisfy the necessary condition as illustrated in Eq. (9) for
the binary case. On the other hand in the multiclass setting, even if we know $\max_y \eta_y = 0.7$, the class probabilities
of the other classes can be *any positive values as long as they are sum to* $0.3$. This makes it more difficult to tune the
hyperparameters as we also showed in Eq. (9) that no values of $\alpha, \beta$ can satisfy both equations in (6). We will improve
the clarity of our writing in Lines 144–158.

(Q2) Relationship to deep learning literature and the reviewer's suggested work.
(A2) To enable the use of deep learning in multiclass classification with rejection is definitely an important future
research direction. We checked Eq. (5) in "Learning Confidence for Out-of-Distribution Detection in Neural Networks"
and found that the proposed objective function looks similar to the separation-based approach but still different. The
rejection criterion is not discussed but instead they have the confidence value. Moreover, they modified the predictions
while training with their Eq.(2) and this mechanism may make the analysis difficult. Without their Eq. (2), with proper
formalization of their Eq. (5) as a surrogate loss, and with proper formalization of the rejection criterion, e.g., $c < 0.5$,
it should be possible to use our condition. Regarding the problem addressed by "On calibration of modern neural
networks", a failure of class probability estimation generally comes from (1) model misspecification, (2) inappropriate
surrogate loss, (3) insufficient optimization, and (4) overfitting. Since our excess risk bound holds for general models in
learning with reject option, our result suggests that the reason (2) is excluded and other reasons become dominant.
As a future work, it would be interesting to explore the possibility of using our results in the separation-based approach
to help design a better method for training two deep neural networks simultaneously for tackling the problem along this
line of research more effectively.

Response to Reviewer#2

(Q3) State the impossibility theorem clearly.
(A3) If the impossibility theorem means the negative result of the separation-based approach, please see (A1).

(Q4) Significance of the theory for the confidence-based approach.
(A4) As pointed out, it is not surprising that the class probabilities can be estimated by using strictly proper composite
losses. However, to the best of our knowledge, derivation of excess risk bounds has different difficulty, particularly
when the multiclass case and/or the reject option are considered. For the OVA loss, even though some parts of the
proof (in Appendix A.1) reduces to the same line as that of Yuan+ [29] for the binary case, other parts needs analysis
characteristic to the multiclass case. Furthermore, for the CE loss, the derivation of an excess risk bound is much more
difficult. This can be seen from not only the fact that the analysis of [9][29] does not apply to the multiclass case, but
also from [19] where the difficulty of CE loss for the multiclass case is actually discussed. Although one may argue that
the proof techniques we used are not complicated, our results are novel and relevant for multiclass classification with
reject option since these surrogate losses are well-known and used in the literature. Apart from our bounds, we are only
aware of the bounds by Ramaswamy+ [20], which focused on other surrogate losses specially designed depending on
the rejection cost.

(Q5) Experimental results are not impressive.
(A5) The purpose of the experiments is not to show the superiority of new methods, but to verify our theoretical findings.
We successfully verify the sub-optimality of the separation-based APC and MPC experimentally as Reviewer#2
mentioned. Furthermore, another important message from the experiments is that CE is a strong baseline in this
problem, which the existing theoretically-grounded work of multiclass learning with rejection could not outperform
(OVA+hinge by Ramaswamy+ [20]). As one of the references from Reviewer#1 suggested that using the confidence
value from CE may not be highly reliable, CE is still the best way to tackle this problem in our experiments. Our
experiments help illustrate that there are still many things to be done to advance the research in this direction, which is
highly relevant for critical tasks where rejection is preferred over misclassification.

[Meta-Review · NeurIPS 2019]

Given the disparity in the reviewers' scores and the authors' confidential comments to the AC, I looked at the paper carefully myself. My impressions of the paper's main contributions are the following: 1. Unified study of confidence-based and rejection-based methods for multiclass classification with a reject option. (Significance: Medium-high) 2. Sufficient and necessary conditions for calibration of rejection-based methods, together with a demonstration that two natural rejection-based surrogates (APC-exp, MPC-exp) do not satisfy the necessary condition and therefore are not calibrated. (Significance: Medium) 3. Derivation of excess risk bounds for confidence-based methods based on class probability estimation. (Significance: Low) 4. Experimental comparisons of the two classes of approaches. (Significance: Medium) In light of this, I would like to recommend a (weak) accept, with the following conditions: 1. Since the paper establishes calibration failure of only the MPC-exp and APC-exp rejection-based surrogates -- and not of all rejection-based surrogates in general -- the authors must re-word claims throughout the paper accordingly, making clear that it remains an open question whether there might be other rejection-based surrogates that could be calibrated (their current results do not appear to conclusively rule out this possibility). I would also encourage them to see if they can establish failure of the MPC-log and APC-log surrogates which are used in their experiments. 2. The authors must make corrections to their citations to Ramaswamy et al. [20] as follows: (a) p.1: "...empirical performance is not convincing" -- needs to be more precise as to what is not convincing (or change language); (b) p.3: "...defined calibration in this problem as follows" -- as far as I remember there is no such definition in Ramaswamy et al. [20]. 3. The authors must include important experimental details such as number of classes K in the main text. All the above changes should be relatively easy to implement. I note that Reviewer 2 recommended rejecting the paper as some of the main claims are currently not stated accurately. However I believe that with the above changes (all of which should be relatively easy to implement), the claims will be accurately supported and the results will be of value to the NeurIPS community; this is why I am recommending a (weak) accept. I discussed this proposal with the reviewers and they did not object to it. I also note that if the paper is indeed accepted, then it is extremely important that the authors implement the changes recommended above.